# Fluid Inclusions and REE Geochemistry of White and Purple Fluorite: Implications for Physico-Chemical Conditions of Mineralization; an Example from the Pinavand F Deposit, Central Iran

Fatemeh Ghaedi [1], Batoul Taghipour [1,*], Alireza K. Somarin [2] and Samaneh Fazli [1]

1 Department of Earth Sciences, Faculty of Sciences, Shiraz University, Shiraz 71454, Iran; ghaedif92@gmail.com (F.G.); fazlisamaneh@yahoo.com (S.F.)

2 Department of Geology, Faculty of Sciences, Brandon University, Brandon, MB R7A 6A9, Canada; somarina@brandonu.ca

* Correspondence: taghipour@shirazu.ac.ir; Tel.: +98-917-409-1677

**Abstract:** The Pinavand fluorite deposit is hosted by lower Cretaceous carbonate rocks in the structural-geological transitional zone of Central Iran. The purple and white fluorite occur, respectively, as early replacement masses and late cross-cutting veins. Both fluorites have different and distinct physicochemical characteristics. The purple fluorite has higher homogenization temperatures of fluid inclusions (170–260 °C) and lower ∑REE (1.6 ppm) and Y (1.3 ppm) than the white variety (90–150 °C, 11.12 ppm, and 21.3 ppm, respectively). All fluorite samples show positive Y anomalies (Y/Y*) in the range of 1.15–3.5. The average values of La/Ho in the purple and white fluorites are 23.1 and 3.4, respectively. The purple fluorite samples have lower Y/Ho values (an average of 63.45) than the white fluorite samples (an average of 87.64). The Tb/Ca ratio in the Pinavand fluorites ranges between 0.0000000348 and 0.00000105, and the Tb/La ratio varies between 0.01 and 0.4; these values suggest that both fluorite types are "hydrothermal" in origin. The purple fluorites have a lower Sr and a negative Eu anomaly. These differences in concentrations and ratios of various REE suggest that the physico–chemical conditions of mineralization changed during fluorite deposition at the Pinavand. These changes correspond to an increase in oxygen fugacity and pH, which occurred during white fluorite mineralization at lower temperatures. The $\delta^{34}S$ values of the Pinavand barite samples (an average of 23.25‰) are similar to those of seawater sulfate in the upper Proterozoic. The $\delta^{34}S$ values of galena range from −0.2‰ to −3.7‰, compatible with bacterial sulfate reduction (BSR). These features are similar to those in the hydrothermal and magmatic deposits.

**Keywords:** fluorite; REE; fluid inclusions; physico–chemical condition; sulfur isotopes; pinavand deposit

## 1. Introduction

Fluorite commonly occurs as an ore or gangue mineral during hydrothermal processes from the beginning of the pegmatitic to the end of the hydrothermal phase (e.g., [1–3]). Fluorite maintains properties of the mineralizing fluid such as rare earth element (REE) and Y pattern; hence, it can be utilized as a reliable geochemical tool to reconstruct the ancient and modern physicochemical parameters of hydrothermal systems [4–8]. REE and Y form complexes with fluoride during the evolution of the F-rich hydrothermal fluids and progressively find enriched in F-bearing solutions [9,10]. As a result, analysis of trace elements, including REE and Y, in fluorite provides essential information about the metal sources, temperature conditions, fluid migration, rock–fluid interactions, and fluid-phase chemical composition [5,8,10–14].

The sediment-hosted fluorite deposits play an important role in Iranian fluorite resources. These deposits have been extensively hosted by dolomites and dolomitized limestones of mainly the middle Cretaceous–Triassic [15] in Alborz (Elika Formation) and

Central Iran zones (Shotori Formation; [16,17]; Figure 1). The distribution pattern of F-rich deposits in Iran has not been fully investigated [17]. The Pinavand deposit is one of the best-known F-rich deposits in the late Cretaceous sedimentary units of Southwestern Central Iran (Figure 1), with an estimated reserve of over 1.5 Mt [18,19]. Although the deposit has been studied previously (e.g., [18,20]) there are ambiguities concerning the effects of special processes in ore formation, their relationship with other geological complexes, and the genesis of the deposit. This study aims to use micro-thermometric data along with the geochemistry of trace elements, particularly REE and Y, in various types of the Pinavand fluorites to unravel some of these ambiguities and enhance our conception of the formation conditions of fluorite deposits. Such understanding is crucial and can help increase the success rate of exploring similar deposits in Iran and other parts of the world.

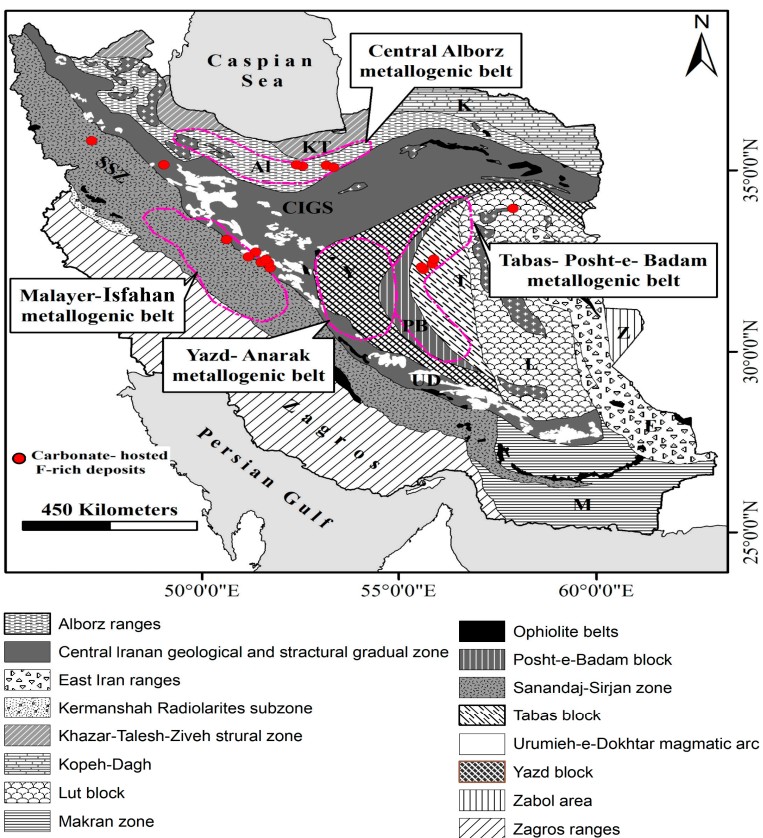

**Figure 1.** Distribution map of carbonate-hosted F-rich deposits in Iran (after [17]).

## 2. Regional Geological Setting

Iran has three main tectonic units: the Alborz Range, the Zagros orogenic belt, and Central Iran [21]. The Zagros orogenic belt formed on the Northern Afro–Arabian plate margin as a result of the Afro–Arabian and Eurasian plates collision following subduction of the Neo–Tethys oceanic crust beneath the Iranian plate during the late Cretaceous to Paleocene [22–26]. This belt is subdivided into four parallel zones with a NW-SE trend: (1) the Urumia–Dokhtar Magmatic Arc (UDMA), (2) the Sanandaj–Sirjan Zone (SSZ), (3) the high Zagros Belt, and (4) the Zagros Fold Belt (ZFB) [21]. Central Iran occurs in the central part of the Iranian plate, between the Zagros and Alborz Ranges [27] (Figure 2a). The Southwestern margin of Central Iran has been intruded by a wide range of late Cretaceous to Miocene magmatic intrusions (related to UDMA) with calc-alkaline to alkaline chemical composition [28,29] (Figure 2). The Central Iran zone is subdivided into several basins and blocks that are separated by boundary faults and geosutures (e.g., [30]). The early Precambrian rock series, as the main component of Central Iran, was deformed during the Katanga orogenic phase. These series are covered by shallow continental to marine

sedimentary rocks of the Precambrian to Triassic [27,31]. The southwest of Central Iran is separated from SSZ by a belt of straight and sloping faults [32], but this border is not distinguished well from other areas due to extensive tertiary and quaternary rock cover, lateral changes of facies, and complex deformations (27). The SSZ is identified by metamorphic rocks related to numerous intrusive masses and extensive Mesozoic volcanism [23,33]. Magmatism and deformation were intense during the Maastrichtian–Paleocene period in Central Iran. The resulted rocks were eroded and then covered by the late Paleocene–Eocene sequences, forming an obvious discontinuity [21]. Intrusion of large plutons into the upper Cretaceous carbonate rocks in the north SSZ and formation of basal conglomerates have occurred during the lower Eocene to middle Eocene [34].

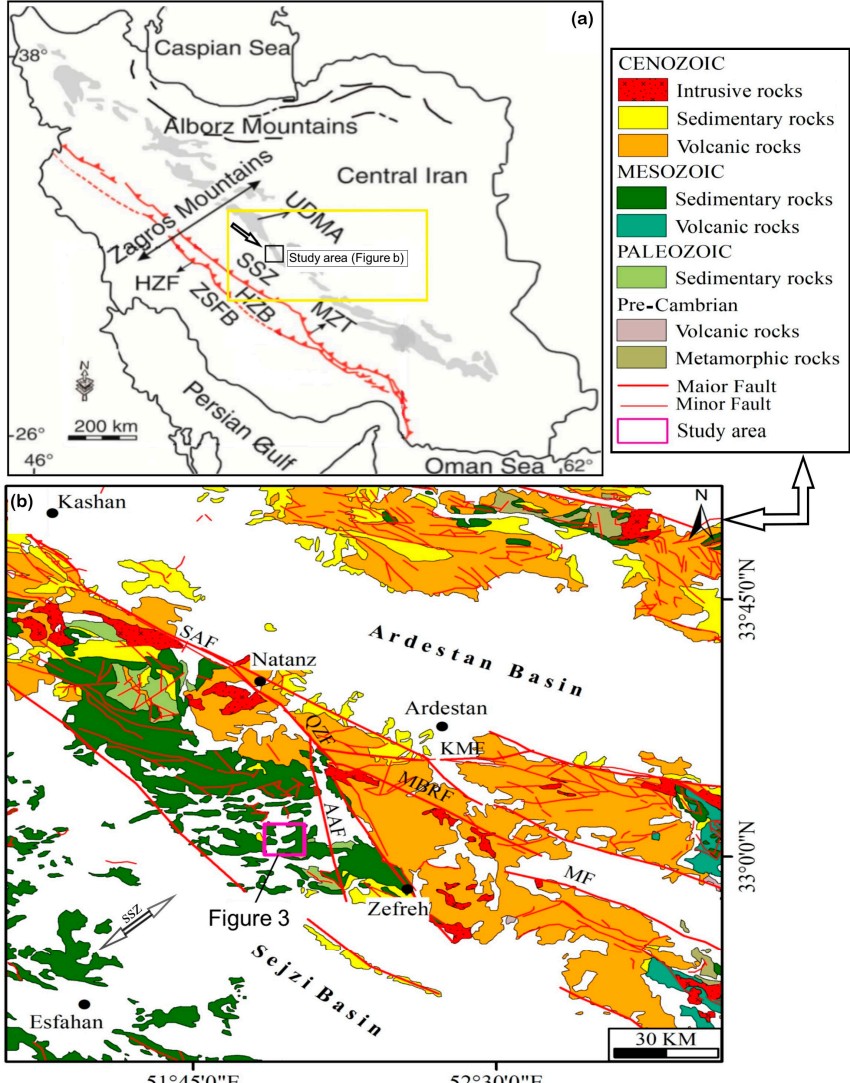

**Figure 2.** Geological maps of Iran showing: (**a**) The main structural zones (after [21]); (**b**) Southwestern part of Central Iran showing the structural location of the Pinavand deposit in relation to the zones, and fault patterns of the central part of the UDMA. Major fault: MBRF = Marbian Rangan; AAF = Abbas Abad; KMF = Kacho Mesghal; MF = Marshenan; SAF = South Ardestan; ZF = Zefreh [35–37]; UDMA = Urumia–Dokhtar Magmatic Assemblage; SSZ = Sanandaj–Sirjan Zone; CIGS = Central Iranian Geological and Structural gradual zone; ZFB = Zagros Fold Belt; HZF = High Zagros Fault; HZB = High Zagros Belt; MZT = Main Zagros Thrust; Ardestan and Sejzi Basins = Sedimentary basins in the Central Iranian zone [31].



## 3. Geology of the Pinavand Area

The Pinavand mine district, 60 km northeast of Isfahan, is located in the structural and geological transition zone of Central Iran (CIGS) and the Sanandaj–Sirjan zone [17,27]. The location of the Pinavand fluorite deposit in CIGS and its proximity to two magmatic zones, UDMA and SSZ, contribute to the complexity of its origin. The fluorite host rocks are the lower Cretaceous calcareous units, including Orbitolina-bearing limestones, silty shales, and sandy limestones (Figure 3) [31].

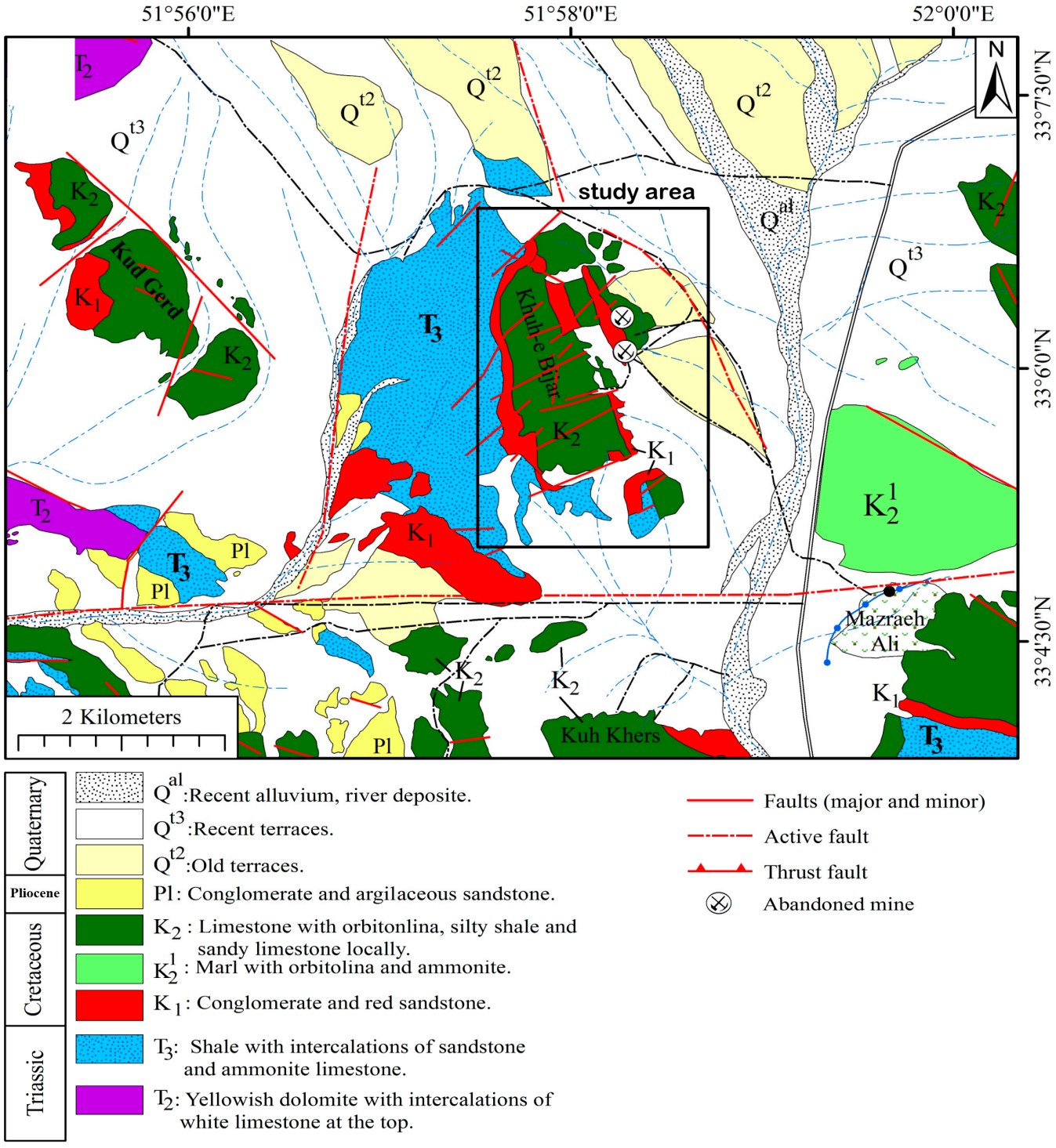

**Figure 3.** Geological map of the Pinavand area.

*Mineralization and Textural Relationship*

The Pinavand ore mineralization was controlled by two geological factors: (1) dolomitic limestones as the carbonate host rock; and (2) the Milajerd–Zefreh fault, which is one of the three strike-slip faults in the region [19]; this fault separates igneous and sedimentary units from each other in the east of the Pinavand region (Figure 2b).

The Pinavand deposit consists of several ore bodies. The main ore body is 2 km long and has an unknown depth. Mineralization occurs as veins, open space filling, and the replacement of carbonate host rocks (Figures 4 and 5). Two types of veins are identified: fluorite-rich veins with 20% to 60% (average 30%) F, and fluorite-barite-galena veins [18]. Fluorite-rich veins are characterized by variable thickness and comb/radial structures and are mainly hosted by Cretaceous limestone (Figure 4d–f). In contrast, fluorite-barite-galena veins have various generations of minerals (fluorite, barite, galena, and quartz) and are not limited to calcareous rocks.

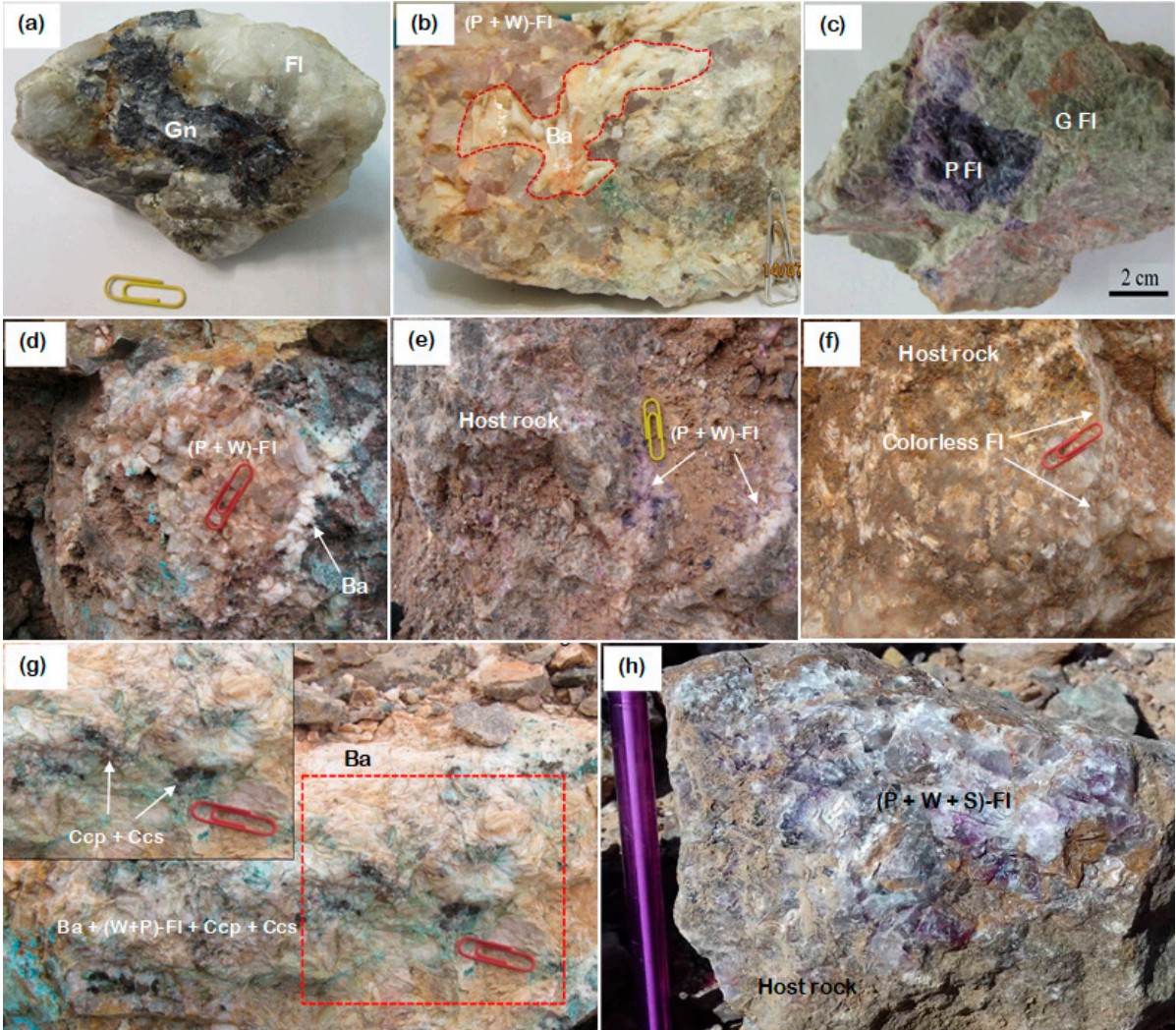

**Figure 4.** Mineralogical and textural features of the Pinavand deposit: (**a**) White massive fluorite associated with galena; (**b**) Purple-fluorite-white fluorite-barite assemblage; (**c**) Open space filling purple and smokey green fluorite; (**d**) Zoned mineralization of purple and white fluorite in the core and barite in the rim; (**e**) Purple and white fluorite with replacement texture in dolomite host rock; (**f**) Dolomite replaced by massive colorless fluorite; (**g**) Radial-fibrous barites with pseudo-acicular texture; (**h**) Association of white, purple, and smoky fluorites; (W Fl = White Fluorite, P Fl = Purple Fluorite, S Fl = Smoky Fluorite, G Fl = Green Fluorite, Gn = Galena, Ba = Barite, Ccp = Chalcopyrite, Ccs = Chalcocite).

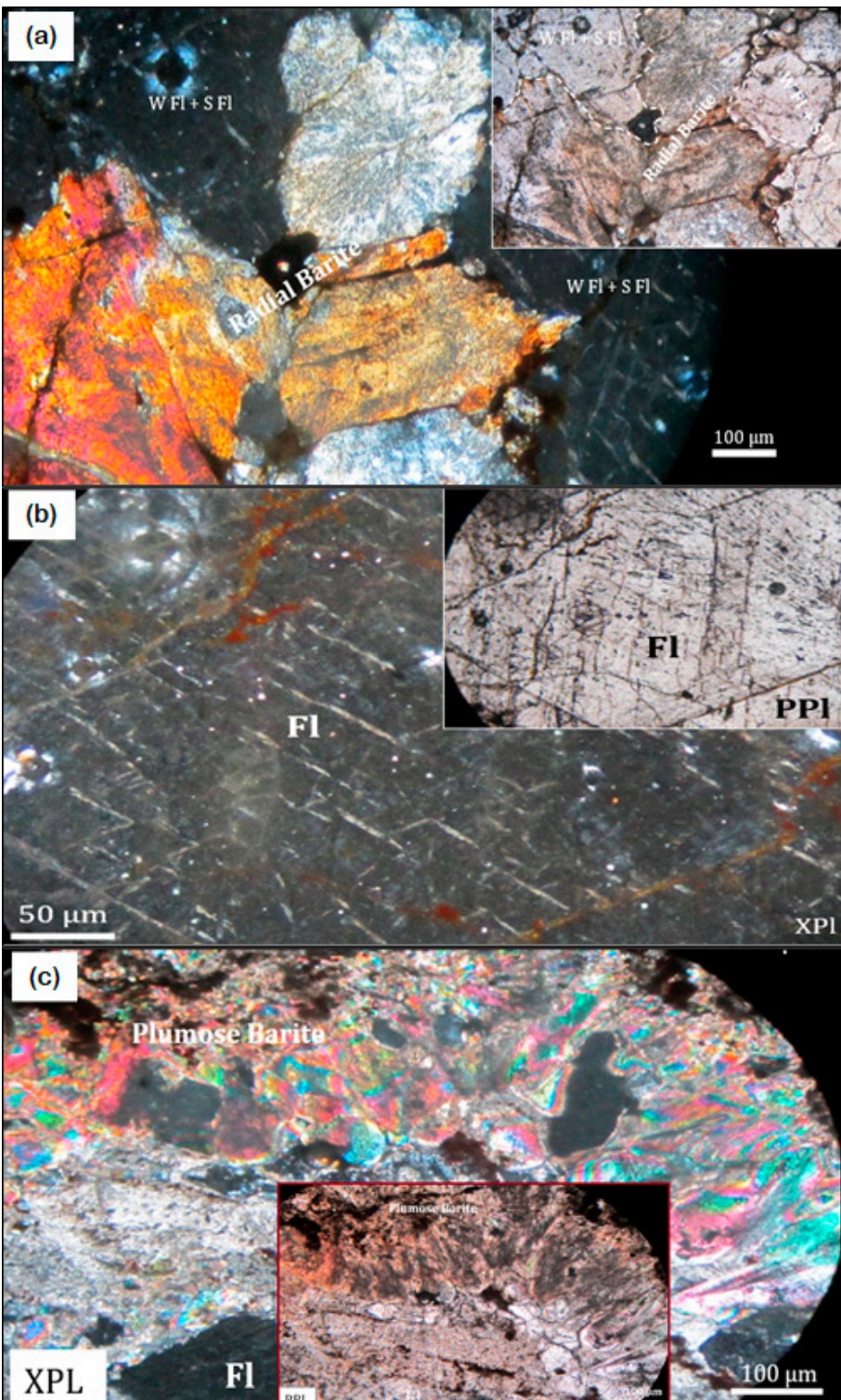

**Figure 5.** Photomicrograph of mineral associations and textures from the Pinavand ore: (**a**) Radial texture of barite with white and smoky fluorite; (**b**) Fluorite; (**c**) Plumose texture of barites associated with purple fluorite; (W + P Fl = White + Purple Fluorite, PPL = Plane Polarized Light, XPL = Cross Polarized Light).

The limestone host rock shows dolomitization and silicification. Silicified zones have fluorite, galena, pyrite, and trace sphalerite. Sulfide minerals are scarce and occur throughout the deposit. Fluorite, as the most common mineral, shows a variety of colors, including white, purple, smoky, green, and colorless (Figure 4).

Silicification occurred before the main fluorite mineralization stage. It formed hydrothermal quartz veins and replacement masses in the carbonate host rocks. Dolomitization is a less common alteration that forms sporadic dolomite in the limestone host rock.

In the Pinavand deposit, the fluorite veins show a zoning where dark to light purple fluorites are gradually replaced by white to smoky or cream fluorites. This indicates that the fluorite mineralization occurred in two stages, including partial or complete dissolution of the older generation (Figure 4h).

The Pinavand fluorite shows a variety of textures, including radial-fibrous, pseudo-acicular, and plumose textures (Figures 4g and 5a,c). These textures were formed due to fluid circulation in the fractures and boiling (mostly in epithermal deposits) [38–40]. These textures are also found in barite, which accompanies purple and white fluorites (Figures 4g and 5a,c).

## 4. Sampling and Analytical Methods

Mineralogy, geochemistry, and micro-thermometry studies were conducted on 27 representative samples collected from the tailings and outcrops of the Pinavand deposit. Chemical analysis of different generations of fluorite was carried out on 8 white and 12 purple fluorite samples in the ACME laboratory, Canada. For this analysis, a calcined or ignited sample (0.2 g) was added to 1.5 g of Lithium Borate Flux ($LiBO_2$), mixed well, fused in a furnace at 1000 °C and then dissolved in 100 mL of 5% $HNO_3$. The contents were analyzed by inductively coupled plasma optical emission spectroscopy (ICP-OES) and inductively coupled plasma mass spectrometry (ICP-MS) following dissolution of the sample by four acid digestions (HF, $HClO_4$, $HNO_3$, and HCl). A 0.25 g sample of rock powder was first digested using hydrofluoric acid (HF), then digested with a mixture of nitric and perchloric acids ($HNO_3$ and $HClO_4$), before being heated in several ramping and holding cycles using precise, programmer-controlled heating that took the samples to incipient dryness. At this stage, each sample was dissolved in aqua regia before being analyzed using ICP-OES and ICP-MS instruments. Analytical uncertainties vary from 0.01 to 0.5 ppm for rare earth elements; from 0.1% to 0.5% for trace elements; and from 0.04% to 0.1% for major elements (Tables 1 and 2).

Microthermometry was carried out on doubly polished wafers of white fluorite using a Linkam THMS600 heating-freezing stage mounted on a Zeiss Axioplan 2 imaging microscope at Lorestan University, Iran. The stage was calibrated in the range −196 °C to +600 °C using n-hexane (melting point of −94.3 °C) and cesium nitrate (melting point of 414 °C). The uncertainty in the heating and freezing measurements is ±0.6 °C and ±0.2 °C, respectively. The fluid inclusion salinity (in wt% NaCl equiv.) in the $H_2O$–NaCl binary system was calculated using Bodnar [41]. The collected purple fluorite samples are cloudy and do not contain measurable fluid inclusions. The purple fluorites previously studied by Qishlaqi [18] (temperature range of 170 °C to 260 °C) were used to compare with the white fluorite data (temperature range of 90 °C to 150 °C; Table 3).

The sulfur isotope composition of sulfide (*n* = 4) and sulfate (*n* = 2) samples was measured at the G.G. Hatch Isotope Laboratory, University of Ottawa, Canada. The $\delta^{34}S$ values were determined by analyzing $SO_2$ formed by the combustion of samples at 1800 °C on an elemental microanalyzer. The released gaseous $SO_2$ was transported by extra-pure helium and then separated by the "trap and purge" method. $SO_2$ gas was transported by helium to a Delta XP isotope ratio mass spectrometer (Thermo Finnigan, Bremen, Germany) through a ConFlo IV interface for $^{34}S$ determination, with an accuracy of better than ±0.2 per mil (‰). The isotope ratios are reported relative to the Canyon Diablo Troilite ($\delta^{34}S$V-CDT) standard (Table 4).

**Table 1.** Major oxide and trace element composition of the Pinavand fluorites.

| Sample No. | Detection | WF-01 | WF-02 | WF-05 | WF-07 | WF-06 | WF-12 | WF-13 | WF-20 | PF-03 | PF-04 | PF-08 | PF-09 | PF-10 | PF-11 | PF-15 | PF-14 | PF-16 | PF-17 | PF-18 | PF-19 |
|---|---|---|---|---|---|---|---|---|---|---|---|---|---|---|---|---|---|---|---|---|---|
| Colors | Limits | White | White | White | White | White | White | White | White | Purple | Purple | Purple | Purple | Purple | Purple | Purple | Purple | Purple | Purple | Purple | Purple |
| $SiO_2$ | wt% | 0.06 | 0.6 | 0.71 | 0.07 | 0.57 | 0.39 | 0.33 | 0.35 | 0.75 | 0.62 | 2.04 | 0.47 | 2.8 | 1.94 | 1.25 | 0.68 | 2.37 | 0.97 | 1.8 | 2 |
| $Al_2O_3$ | wt% | <0.01 | <0.01 | <0.01 | <0.01 | <0.01 | <0.01 | <0.01 | <0.01 | <0.01 | <0.01 | 0.03 | <0.01 | <0.01 | 0.02 | <0.02 | <0.01 | <0.02 | <0.02 | <0.02 | <0.02 |
| $Fe_2O_3$ | wt% | <0.04 | 0.13 | <0.04 | <0.04 | <0.04 | <0.04 | 0.13 | 0.07 | 0.06 | 0.11 | 0.13 | 0.24 | 0.15 | 0.05 | 0.18 | 0.08 | 0.1 | 0.14 | 0.11 | 0.1 |
| MgO | wt% | <0.01 | <0.01 | <0.01 | <0.01 | <0.01 | <0.01 | <0.01 | <0.01 | <0.01 | <0.01 | 0.01 | <0.01 | <0.01 | 0.02 | 0.01 | <0.01 | 0.02 | 0.01 | <0.01 | 0.01 |
| $Na_2O$ | wt% | 0.02 | 0.02 | 0.02 | 0.02 | 0.02 | 0.02 | 0.02 | 0.02 | 0.02 | 0.01 | 0.02 | 0.02 | 0.02 | 0.03 | 0.02 | 0.01 | 0.02 | 0.01 | 0.02 | 0.03 |
| $K_2O$ | wt% | <0.01 | <0.01 | <0.01 | <0.01 | <0.01 | <0.01 | <0.01 | <0.01 | <0.01 | <0.01 | 0.01 | <0.01 | <0.01 | 0.02 | 0.01 | <0.01 | 0.02 | <0.01 | <0.01 | 0.02 |
| $Cr_2O_3$ | wt% | <0.002 | <0.002 | <0.002 | <0.002 | <0.002 | <0.002 | <0.002 | <0.002 | 0.003 | 0.003 | <0.002 | <0.002 | <0.002 | <0.002 | <0.002 | <0.002 | 0.003 | <0.002 | 0.002 | <0.002 |
| F | wt% | 27.78 | 27.68 | 25.77 | 28.61 | 26.2 | 27.2 | 27.7 | 28.5 | 24.84 | 29.11 | 23.64 | 26.66 | 29.26 | 24.79 | 25.13 | 27 | 27.01 | 26.1 | 27 | 24.5 |
| Ba | 1 ppm | 37 | 61 | 67 | 6727 | 1650 | 3397 | 49 | 3420 | 8 | 15 | 99 | 57 | 415 | 74 | 78 | 11.5 | 233.5 | 45.2 | 210.5 | 85.8 |
| Co | 0.2 ppm | 3.9 | 11.4 | 2.6 | 5.1 | 3.3 | 3.85 | 7.65 | 8.3 | 14.2 | 13.8 | 9.1 | 6.6 | 4.5 | 8.9 | 7.8 | 14 | 6.7 | 11 | 9.4 | 9 |
| Mo | 0.1 ppm | 0.2 | 0.2 | 0.2 | 0.2 | 0.2 | 0.2 | 0.2 | 0.2 | 0.2 | I.S. | 0.2 | 0.2 | 0.3 | 0.2 | 0.2 | 0.2 | 0.2 | 0.2 | 0.2 | 0.2 |
| Pb | 0.1 ppm | 42 | 10.1 | 20.2 | 33 | 24 | 26.6 | 26.05 | 22 | 11.6 | I.S. | 48.4 | 5.8 | 128.4 | 28.8 | 27.1 | 11.6 | 78.6 | 20.1 | 75.2 | 39.5 |
| Ni | 0.1 ppm | <0.1 | <0.1 | <0.1 | <0.1 | <0.1 | <0.1 | <0.1 | <0.1 | 1.1 | I.S. | 0.2 | 0.1 | 1.5 | <0.1 | 0.1 | 1.1 | 1.5 | 0.7 | 1.2 | 0.2 |
| Au | 0.5 ppm | 1.1 | <0.5 | <0.5 | 1.6 | 0.9 | 1.6 | 1.1 | 1.6 | 3.4 | I.S. | <0.5 | <0.5 | <0.5 | <0.5 | <0.5 | 3.4 | <0.5 | <0.5 | 2.2 | <0.5 |
| Th | 0.2 ppm | 0.8 | 0.9 | 1 | 1 | 0.9 | 1 | 0.8 | 1 | 0.2 | 0.2 | 0.2 | 0.2 | 0.2 | 0.2 | 0.2 | 0.2 | 0.2 | 0.2 | 0.2 | 0.2 |

**Table 2.** REE and Ca concentration in white (WF) fluorite, purple (PF) fluorite, and carbonate host rock (LST) of the Pinavand deposit.

| Sample No. | WF-01 | WF-02 | WF-05 | WF-07 | WF-06 | WF-12 | WF-13 | WF-20 | PF-03 | PF-04 | PF-08 | PF-09 | PF-10 | PF-11 | PF-14 | PF-15 | PF-16 | PF-17 | PF-18 | PF-19 | LSt-16 | LSt-17 |
|---|---|---|---|---|---|---|---|---|---|---|---|---|---|---|---|---|---|---|---|---|---|---|
| Colors | White | White | White | White | White | White | White | White | Purple | Purple | Purple | Purple | Purple | Purple | Purple | Purple | Purple | Purple | Purple | Purple | Limestone | Limestone |
| Ca% | 51.18 | 50.91 | 51.98 | 50.56 | 51.63 | 51.27 | 51.04 | 50.74 | 52.61 | 50.09 | 51.36 | 51.03 | 50.53 | 50.47 | 51.62 | 51.2 | 50.71 | 51.32 | 52 | 51.2 | 17.83 | 17.26 |
| La | 1.1 | 0.5 | 0.9 | 0.9 | 0.9 | 0.9 | 0.8 | 0.7 | 0.4 | 0.3 | 0.7 | 0.5 | 0.5 | 0.3 | 0.3 | 0.6 | 0.4 | 0.5 | 0.4 | 0.5 | 19.3 | 17.11 |
| Ce | 1.8 | 0.8 | 2.2 | 1.7 | 2.2 | 2 | 1.2 | 1.1 | 0.5 | 0.3 | 0.4 | 0.3 | 0.6 | 0.3 | 0.4 | 0.3 | 0.4 | 0.4 | 0.5 | 0.3 | 31.34 | 27.72 |
| Pr | 0.29 | 0.14 | 0.36 | 0.35 | 0.35 | 0.35 | 0.21 | 0.24 | 0.06 | 0.03 | 0.04 | 0.04 | 0.07 | 0.04 | 0.05 | 0.04 | 0.05 | 0.04 | 0.6 | 0.04 | 4.86 | 3.98 |
| Nd | 1.9 | 1 | 2.7 | 2.6 | 2.7 | 2.65 | 1.45 | 1.8 | 0.2 | 0.2 | 0.2 | 0.2 | 0.2 | 0.2 | 0.2 | 0.2 | 0.2 | 0.2 | 0.2 | 0.2 | 21.3 | 18.65 |
| Sm | 1.19 | 0.76 | 1.18 | 1.13 | 1.2 | 1.15 | 0.98 | 0.94 | 0.04 | 0.05 | 0.04 | 0.04 | 0.05 | 0.05 | 0.04 | 0.04 | 0.05 | 0.05 | 0.05 | 0.04 | 3.2 | 2.4 |
| Eu | 0.6 | 0.34 | 0.76 | 0.54 | 0.7 | 0.65 | 0.47 | 0.4 | 0.02 | 0.02 | 0.02 | 0.02 | 0.02 | 0.02 | 0.02 | 0.02 | 0.02 | 0.02 | 0.02 | 0.02 | 1.5 | 1.1 |
| Gd | 1.92 | 1.32 | 2.22 | 2.23 | 2.2 | 2.22 | 1.62 | 1.8 | 0.16 | 0.14 | 0.2 | 0.13 | 0.15 | 0.13 | 0.15 | 0.17 | 0.14 | 0.16 | 0.15 | 0.16 | 4.45 | 3.52 |
| Tb | 0.24 | 0.21 | 0.29 | 0.3 | 0.3 | 0.29 | 0.22 | 0.25 | 0.02 | 0.02 | 0.02 | 0.02 | 0.02 | 0.02 | 0.02 | 0.02 | 0.02 | 0.02 | 0.02 | 0.02 | 0.8 | 0.55 |
| Dy | 1.3 | 1.34 | 1.3 | 1.57 | 1.4 | 1.43 | 1.32 | 1.4 | 0.08 | 0.08 | 0.11 | 0.08 | 0.1 | 0.12 | 0.08 | 0.09 | 0.11 | 0.08 | 0.09 | 0.11 | 4.21 | 3.02 |
| Ho | 0.2 | 0.23 | 0.26 | 0.28 | 0.26 | 0.27 | 0.21 | 0.25 | 0.02 | 0.02 | 0.02 | 0.02 | 0.02 | 0.02 | 0.02 | 0.02 | 0.02 | 0.02 | 0.02 | 0.02 | 0.78 | 0.69 |
| Er | 0.43 | 0.58 | 0.58 | 0.6 | 0.58 | 0.59 | 0.5 | 0.6 | 0.06 | 0.05 | 0.06 | 0.06 | 0.05 | 0.05 | 0.05 | 0.06 | 0.05 | 0.06 | 0.05 | 0.5 | 2.26 | 1.91 |
| Tm | 0.05 | 0.06 | 0.05 | 0.06 | 0.05 | 0.05 | 0.05 | 0.06 | 0.01 | 0.01 | 0.01 | 0.01 | 0.01 | 0.01 | 0.01 | 0.01 | 0.01 | 0.01 | 0.01 | 0.01 | 0.39 | 0.31 |
| Yb | 0.22 | 0.26 | 0.28 | 0.3 | 0.28 | 0.29 | 0.24 | 0.28 | 0.07 | 0.07 | 0.07 | 0.07 | 0.07 | 0.07 | 0.07 | 0.07 | 0.07 | 0.07 | 0.07 | 0.07 | 0.84 | 0.75 |
| Lu | 0.03 | 0.03 | 0.04 | 0.03 | 0.03 | 0.03 | 0.03 | 0.03 | 0.01 | 0.01 | 0.01 | 0.01 | 0.01 | 0.01 | 0.01 | 0.01 | 0.01 | 0.01 | 0.01 | 0.01 | 0.34 | 0.34 |

**Table 2.** *Cont.*

| Sample No. | WF-01 | WF-02 | WF-05 | WF-07 | WF-06 | WF-12 | WF-13 | WF-20 | PF-03 | PF-04 | PF-08 | PF-09 | PF-10 | PF-11 | PF-14 | PF-15 | PF-16 | PF-17 | PF-18 | PF-19 | LSt-16 | LSt-17 |
|---|---|---|---|---|---|---|---|---|---|---|---|---|---|---|---|---|---|---|---|---|---|---|
| Colors | White | White | White | White | White | White | White | White | Purple | Purple | Purple | Purple | Purple | Purple | Purple | Purple | Purple | Purple | Purple | Purple | Limestone | Limestone |
| Y | 19.4 | 17.1 | 23.7 | 25.1 | 24.05 | 24.4 | 18.25 | 21.1 | 1.7 | 1.3 | 1.2 | 1.5 | 0.5 | 1.4 | 1.5 | 1.4 | 0.9 | 1.42 | 1.1 | 1.3 | | |
| ∑REE | 11.3 | 7.4 | 13.2 | 12.6 | 13.05 | 12.9 | 9.3 | 10 | 2.1 | 1.2 | 1.8 | 1.2 | 2 | 1.3 | 1.6 | 1.5 | 1.7 | 1.6 | 2.05 | 1.5 | 95.57 | 82.05 |
| (La/Yb) n | 2.23 | 1.3 | 2.17 | 2.02 | 2.14 | 2.09 | 1.76 | 1.7 | 3.8 | 2.9 | 6.7 | 4.8 | 4.8 | 2.9 | 3.3 | 5.75 | 3.8 | 4.6 | 4.3 | 4.8 | 14.12 | 14.02 |
| (Tb/Yb) n | 4.85 | 3.6 | 4.61 | 4.45 | 4.6 | 4.53 | 4.23 | 4.02 | 1.7 | 1.8 | 1.7 | 1.6 | 1.7 | 1.8 | 1.75 | 1.7 | 1.7 | 1.6 | 1.6 | 1.8 | 0.42 | 0.33 |
| (Tb/La) n | 1.44 | 2.76 | 2.12 | 2.2 | 2.14 | 2.16 | 2.1 | 2.5 | 0.32 | 0.44 | 0.19 | 0.26 | 0.26 | 0.44 | 0.4 | 0.23 | 0.35 | 0.31 | 0.29 | 0.31 | 0.03 | 0.02 |
| La/Ho | 5.5 | 2.17 | 3.46 | 3.21 | 3.4 | 3.33 | 3.83 | 2.7 | 20 | 15 | 35 | 25 | 25 | 15 | 20 | 31 | 20 | 23.8 | 22.4 | 25 | 24.74 | 24.79 |
| Ce/Ce* | 0.77 | 0.38 | 0.85 | 0.73 | 0.85 | 0.85 | 0.66 | 0.64 | 0.7 | 0.76 | 0.57 | 0.51 | 0.77 | 0.66 | 0.73 | 0.54 | 0.71 | 0.63 | 0.72 | 0.61 | 0.75 | 0.78 |
| Eu/Eu* | 1.22 | 1.05 | 1.45 | 1.05 | 1.35 | 1.25 | 1.13 | 1.05 | 0.77 | 0.74 | 0.73 | 0.86 | 0.8 | 0.8 | 0.78 | 0.8 | 0.8 | 0.78 | 0.8 | 0.75 | 1.21 | 1.15 |
| Y/Y* | 3.26 | 2.64 | 3.5 | 3.25 | 3.43 | 3.37 | 2.95 | 2.8 | 2.66 | 2.78 | 2.19 | 3.21 | 1.15 | 2.44 | 2.7 | 2.8 | 1.8 | 2.7 | 1.9 | 2.3 | | |
| Y/Ho | 97 | 74.35 | 91.15 | 89.64 | 90.8 | 90.39 | 85.67 | 82.1 | 85 | 65 | 60 | 75 | 25 | 70 | 73.7 | 67.5 | 47.5 | 71.25 | 56 | 65.5 | | |

$Eu/Eu* = Eu_N / \sqrt{[(Sm_N \times Gd_N)/2]}$, $Ce/Ce* = Ce_N / \sqrt{[(La_N \times Pr_N)/2]}$, $Y/Y* = Y_N / \sqrt{[(Dy_N \times Ho_N)/2]}$.

**Table 3.** Summary of the micro-thermometric data of fluid inclusions in the Pinavand fluorite.

| Sample No. | Host Mineral | Number of Measurements | Th (°C) | Tmice (°C) | Salinity (wt% NaCl eq.) | Density (g/cm³) | Size (μm) | Data Sources |
|---|---|---|---|---|---|---|---|---|
| P6 | White Fluorite | 14 | 95.6 to 187 | −5.7 to −11.9 | 8.8 to 15.9 | 0.96 to 1.07 | 6.7 to 26.3 | This Study |
| P7 | White Fluorite | 16 | 95.3 to 223.1 | −1.6 to −6.9 | 2.6 to 10.3 | 0.91 to 1.01 | 6.4 to 15.4 | |
| P42 | White Fluorite | 10 | 83.3 to 141.7 | −12.3 to −17.3 | 16.2 to 20.4 | 1.05 to 1.10 | 4.2 to 20.5 | |
| | Purple Fluorite | 40 | 130 to 270 | | 2.5 to 36 | | 5–20 | [18] |
| | Fluorite (white and Purple) | 19 | 75 to 189 | −0.2 to −14.8 | 0.3 to 18.6 | 0.96 to 1.09 | 5 to 20 | [20] |

Th: homogenization temperature; Tmice: temperature for final ice melting.

**Table 4.** Sulfur isotope data of galena and barite samples of the Pinavand deposit.

| Sample No. | Mineral | $\delta^{34}$S (‰ VCDT) |
|---|---|---|
| PS1 | Galena | −3.7 |
| PS2 | Galena | −0.3 |
| PS3 | Galena | −0.2 |
| PS4 | Galena | −0.6 |
| PB2 | Barite | 21.1 |
| PB3 | Barite | 25.4 |
| Average PS | Galena | −1.2 |
| Average PB | Barite | 23.25 |

## 5. Results

### 5.1. Geochemistry

The purple fluorite has higher $SiO_2$ (0.47%–2.8%) than the white fluorite (0.06%–0.71%). Concentrations of MgO, $K_2O$, and $Al_2O_3$ in both fluorite types are very low (Table 1). The Ca content varies from 50.09% to 52.61% in purple fluorite and from 50.56% to 51.98% in white fluorites (Table 2). The thorium content of the white fluorites (average 0.9 ppm) is higher than that of the purple fluorites (average 0.2 ppm). Concentrations of Ni and, to some extent, Co in the purple fluorites are higher than those in the white fluorites (Table 1).

White and purple fluorites have different REE concentrations (Table 2). Total REE (∑REE) for the purple fluorite ranges from 1.2 to 2.1 ppm (average 1.6 ppm); the white fluorites have 7.4–13.2 ppm (average 11.12 ppm) ∑REE. The concentration of Y in white and purple fluorites is 17.1–25.1 ppm and 0.5–1.7 ppm, respectively (Table 2). It appears that the color variation of fluorites may be related to the variation in Y and ∑REE content [42,43].

### 5.2. Fluid Inclusion Microthermometry

All measured fluid inclusions occurred either as isolated inclusions or along growth zones in fluorite and are considered primary [44,45]. Their size ranges from 4 to 26 μm, and they have spherical, oval, elongate, and irregular shapes. These inclusions are single-phase and two-phase, containing liquid $H_2O$ and vapor at room temperature (Figure 6a,b), but they range from vapor-rich to liquid-rich varieties. Coexisting liquid-rich and vapor-rich fluid inclusions contained in the Pinavand fluid inclusion assemblage (Figure 6c) suggest boiling of fluids [46–48] (Figure 6c). Homogenization temperatures of the fluid inclusions in the Pinavand white and purple fluorites range from 90 °C to 150 °C and 170 °C to 270 °C, respectively (Table 3); as no pressure correction was applied to the Th values, these temperatures reflect the minimum temperature of trapping of the fluids. The salinity of fluid inclusions in purple fluorites (2.5–36 wt% NaCl equiv.) is higher than that in white fluorites (2.6–20.4 wt% NaCl equiv.) (Table 3). The calculated density varies between 0.91 and 1.10 g/cm³.

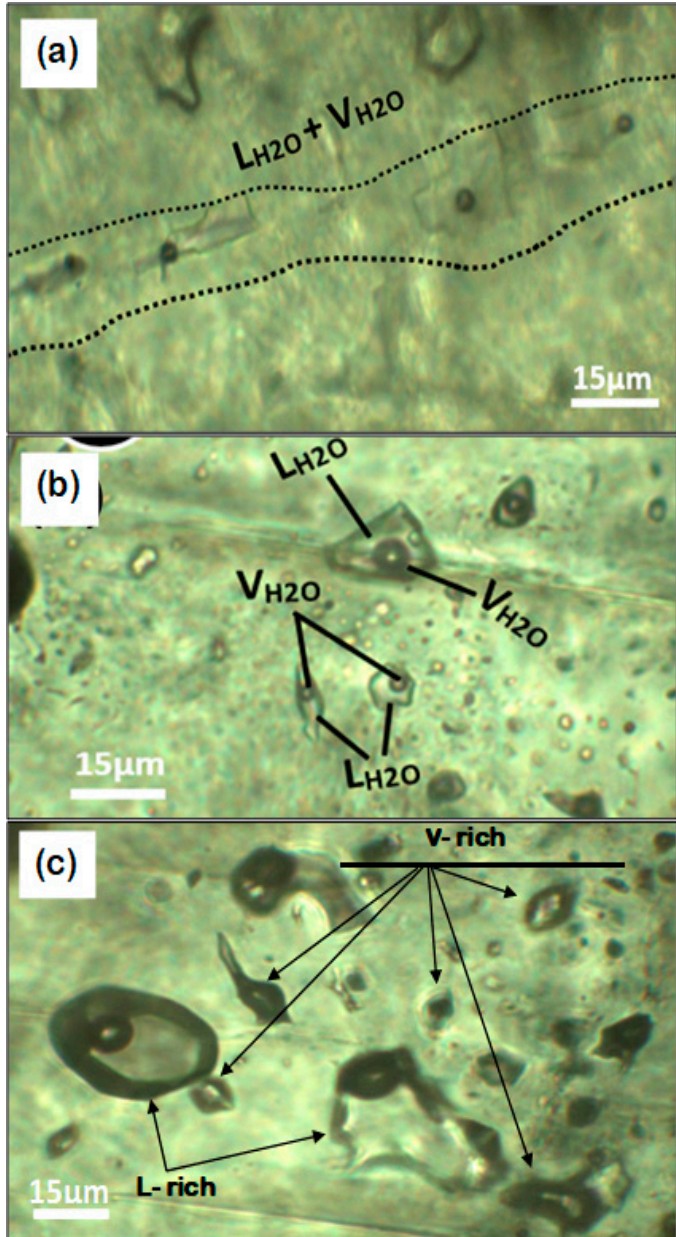

**Figure 6.** Photomicrographs of fluid inclusions in the Pinavand white fluorite. (**a**,**b**) Two-phase; (**c**) Fluid inclusion assemblage consisting of coexisting liquid-rich and vapor-rich varieties.

## 6. Discussion

### 6.1. Tb/Ca and Tb/La Ratios

The Tb/La versus Tb/Ca plot is used to determine the physicochemical conditions of fluorite formation, their degree of fractionation from hydrothermal fluid, and to classify fluorites into sedimentary, hydrothermal, and pegmatitic types [49,50]. The Tb/Ca ratios in the Pinavand fluorites vary between 0.0000000348 and 0.00000105. The Tb/La ratio ranges between 0.01 and 0.4. Low values of the Tb/La ratio suggest the formation of fluorites from LREE-enriched fluids in the early stages of mineralization [14,50–52]. The white and purple fluorite samples plot in two different areas within the "Hydrothermal" field (Figure 7), indicating a prominent role of hydrothermal activity for concentrating REEs in these fluorites [14,53]. This may also indicate that both fluorite types were precipitated from two chemically distinct hydrothermal solutions. Two samples of purple fluorite with higher Tb/La ratios plot close to the sedimentary field (Figure 7), which could be due to the partial reaction of the hydrothermal fluid with the sedimentary host rock [50,54].

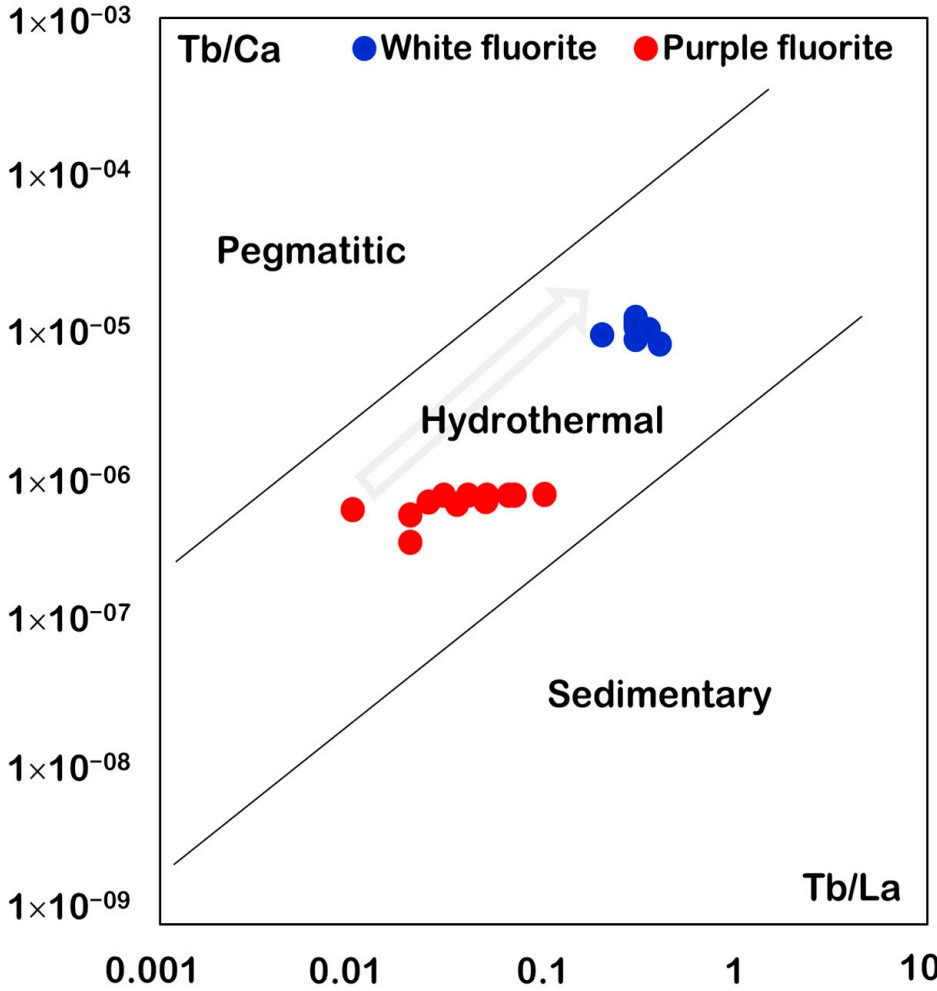

**Figure 7.** Tb/Ca versus Tb/La diagram showing the type of the Pinavand fluorite (boundaries after [50]).

### 6.2. Y–Ho Fractionation

The average Y in purple and white fluorite samples is 1.3 and 21.64 ppm, respectively. Both fluorite types show positive Y anomalies in the range of 1.15 to 3.5 (average 2.69) (Table 2). These positive values suggest strong fractionation of Y-Ho in the Pinavand hydrothermal system. Y-Ho fractionation depends on the composition and migration of the fluid, which may be unrelated to its source [4]; yttrium enrichment mainly depends on the presence of fluoride complexes [55], and Y-F complexes are more stable than Ho-F complexes [56]; hence, it is expected that the Y/Ho ratio increases in fluorine-rich solutions (Figure 8; [57]).

Hydrothermal fluorites are identified by their high Y/Ho ratios (average between 35 and 250; [4]). This ratio in the studied fluorite samples ranges from 25 to 97 (average 80.21) (Table 2); this range is much higher than the chondrite Y/Ho ratio [58] but overlaps with Y/Ho values in hydrothermal fluorites (Figure 8). This may suggest that the Pinavand fluorite has a mixed seawater-igneous source. Similarly, high Y/Ho ratios were found in the Bobrynets, Turkey, and the Tumen fluorite deposits, China. In these deposits, fluorite mineralization was the result of interaction between magmatic fluids and carbonate rocks [14,59].

The average La/Ho ratios in purple and white fluorites are 23.1 and 3.45, respectively. The purple fluorite samples have a lower Y/Ho (average of 63.45) than the white fluorite samples (average of 87.63). The wide La/Ho range and limited Y/Ho range of the Pinavand fluorites (Figure 9) could be due to partial loss of a LREE-rich phase during recrystallization of fluorites [4].

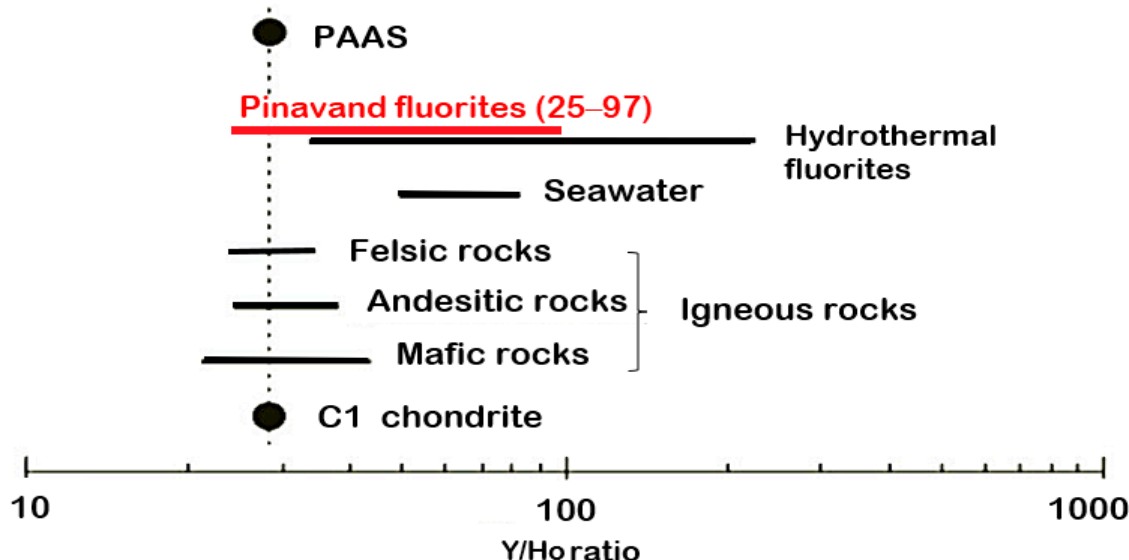

**Figure 8.** Y/Ho ratio in the Pinavand fluorite deposit compared with other geological environments (after [4]). PAAS = Post-Archean Australian Shale.

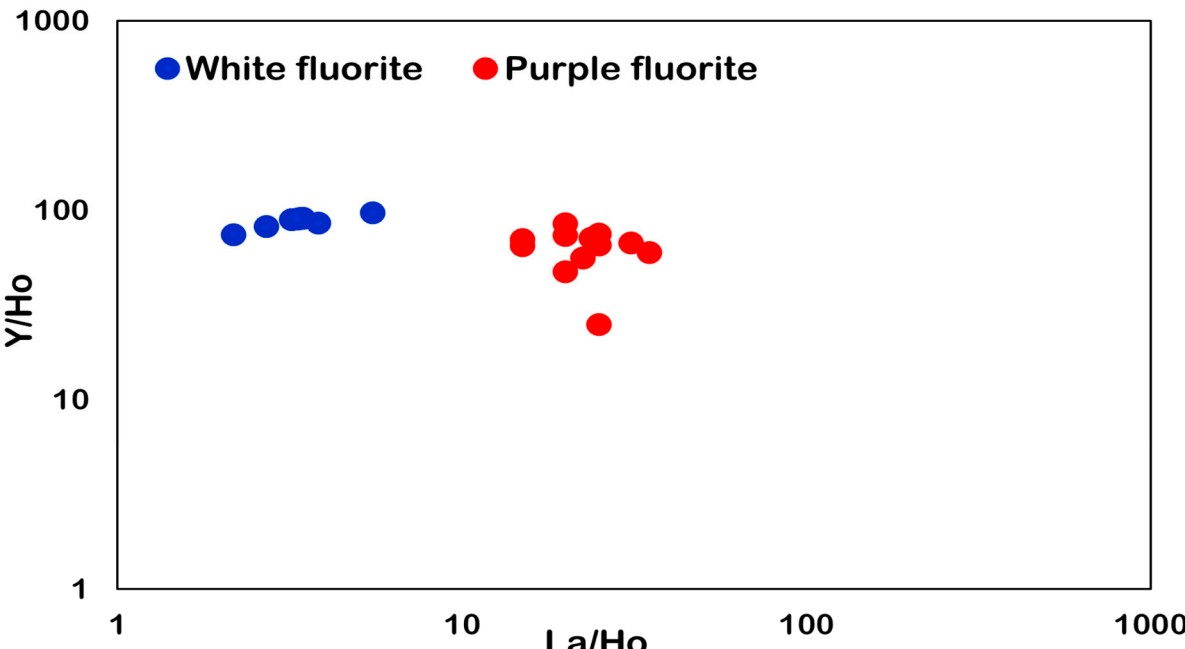

**Figure 9.** Y/Ho–La/Ho diagram for white and purple fluorites of the Pinavand deposit.

*6.3. Eu and Ce Anomalies*

Eu and Ce anomalies have been used as indicators of physicochemical conditions of hydrothermal fluids such as temperature, pH, and oxygen fugacity $(fO_2)$ [14,59,60]. The Eu/Eu* ratio ranges from 0.7 to 0.86 in purple fluorites and shows a negative anomaly (i.e., <1). This ratio in white fluorites varies from 1.05 to 1.45 and shows a positive anomaly (i.e., >1; Table 2). Such positive or negative anomalies in fluorite samples can be related to one or more factors such as changes in pH, $fO_2$, and temperature [14,59,61,62]. The negative Eu anomaly reflects the presence of $Eu^{2+}$ (instead of $Eu^{3+}$) in the hydrothermal solution during purple fluorite deposition [14,53]. As a result, at a temperature above 200 °C, due to the radius difference, $Ca^{2+}$ has not been replaced by $Eu^{2+}$. Therefore, Eu depletion is observed in fluorites [5,13,52,63]. Such a solution precipitates fluorite as the temperature decreases or $fO_2$ increases [5,61]. Fluid inclusion data in the Pinavand area indicates

the purple fluorites were formed at high temperatures (i.e., >200 °C). Co-precipitation of fluorite with other Eu-enriched or depleted minerals may also cause positive or negative anomalies [59]. The co-existence of Eu-depleted and Eu-enriched fluorites reflects the role of two mixed fluids with different temperatures and Eu concentrations [12].

All fluorite samples of the Pinavand deposit show a negative Ce anomaly (i.e., Ce/Ce* < 1) from 0.38 to 0.85. The consistent negative Ce anomaly in the Pinavand fluorites may be due to: (1) oxidation of the solutions at the source [52] causing $Ce^{3+}$ oxidation and $Ce^{4+}$ immobilization [43,52]; (2) involvement of a reduced fluid, which is supported by the presence of sulfide minerals at Pinavand [55] (Figure 4a); (3) formation of hydroxide complexation [63]; the hydroxide complex formed by Ce is more stable than other REEs [64], which causes Ce to remain in the fluid and, consequently, a negative Ce anomaly is observed in the precipitants [59].

### 6.4. REE Enrichment

The enrichment process and the degree of separation of LREE from HREE can be used as tools to determine the primary and secondary generations of fluorite. This is achieved through different ratios of REE, such as La/Yb and Tb/Yb [5].

The Pinavand purple fluorites are more enriched in La, whereas the white fluorites have a higher enrichment in Tb (Figure 10 and Table 2). As a result, the Pinavand white fluorites have higher (Tb/Yb)n and lower (La/Yb)n than the purple varieties (Figure 10). The white varieties are similar to those in the New Mexico deposit, whereas the purple ones are more similar to those in the Lordsburg and Akdagmadeni deposits [1,2,14,53].

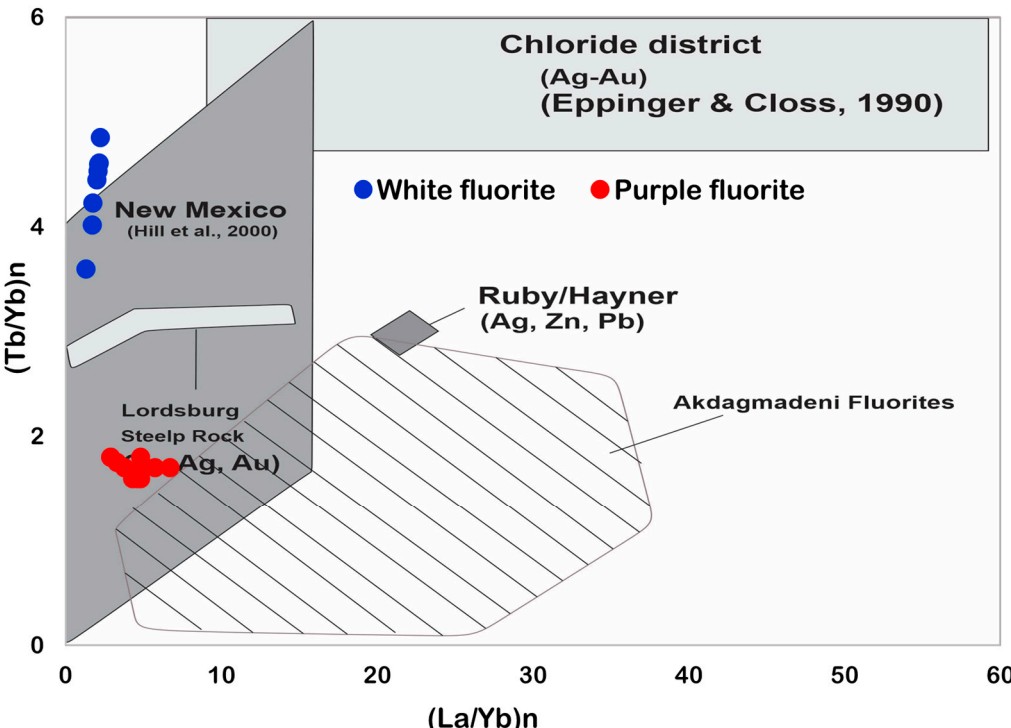

**Figure 10.** (Tb/Yb)n versus (La/Yb)n diagram showing similarity of the Pinavand fluorites to some known deposits [2].

The (La/Yb)n–(Eu/Eu*)n diagram shows that the purple fluorites have LREE enrichment and a negative Eu anomaly, whereas the white fluorites have higher HREE enrichment and a positive Eu anomaly (Figure 11). Both fluorite types of the Pinavand deposit plot in or near the Hansen and Chise vein deposits, which are intermediate- to high-temperature hydrothermal barren fluorite deposits [14,53].

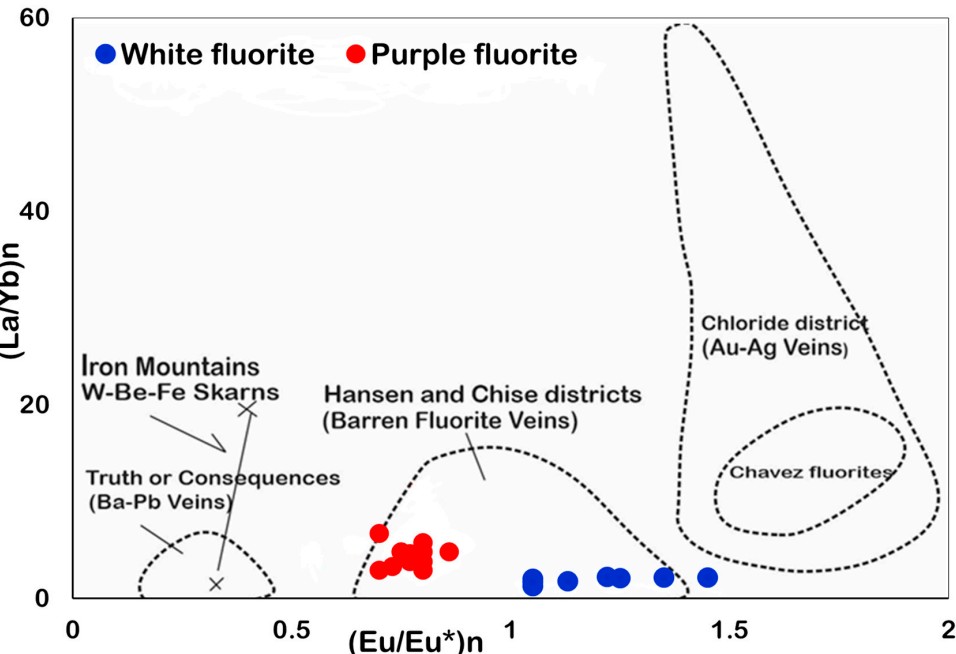

**Figure 11.** (La/Yb)n versus (Eu/Eu*)n diagram showing that the Pinavand fluorites plot at or near the barren fluorite veins field.

The Sr–(Eu/Eu*)n binary diagram shows that the purple fluorites have a lower content of Sr and a negative Eu anomaly, whereas the white fluorites show a positive Eu anomaly and a higher content of Sr (Figure 12). In terms of Sr and (Eu/Eu*)n ratio, the Pinavand purple fluorites are similar to the Hansen and Chise hydrothermal deposits, and the Pinavand white fluorites are similar to the Buyukcal magmatic-hydrothermal fluorites.

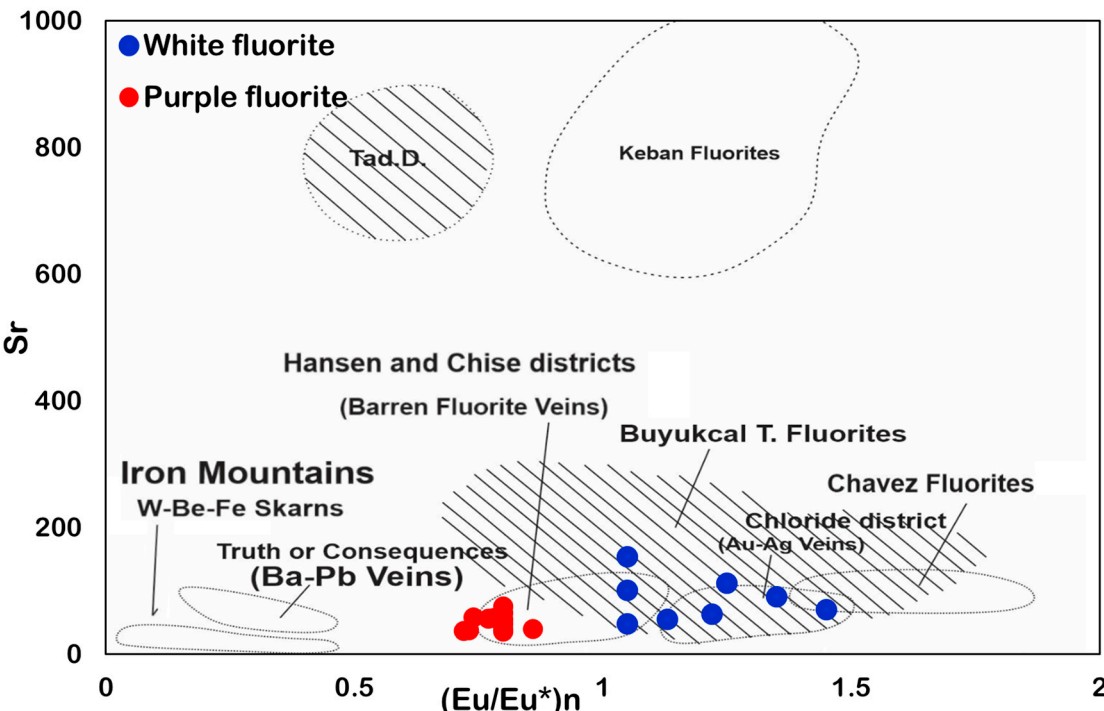

**Figure 12.** Sr versus (Eu/Eu*)n diagram comparing the Pinavand fluorites with other fluorite deposits (fields after [53]).

### 6.5. Evolution of REEs

In hydrothermal systems, pH and the chemical composition of the solutions are two key factors controlling the $\sum$REE content [5,59,65]. For example, $\Sigma$REE in the fluid is increased by decreasing the pH due to the involvement of the F-rich acidic fluids [66]. The average $\Sigma$REE in purple and white fluorites of the Pinavand is 1.6 ppm and 11.22 ppm, respectively. This indicates an increase in $\sum$REE content from purple to white fluorites, reflecting a change in pH or composition of the mineralizing fluid. It seems that the initial fluid was acidic and rich in HF and F in the Pinavand hydrothermal system (reaction I; [67]). As a result, the hydrothermal fluid could likely carry the $REE^{3+}$ in solution as REE-F complexes [4,68]. In this acidic fluid, the activity of $S^{2-}$ was low as it was converted to $HS^-$ (reaction II; [69]). The presence of $S^{2-}$ or $HS^-$ reduces the environment, and consequently, $REE^{3+}$ tends to remain in the fluid; however, $Eu^{3+}$ is reduced to $Eu^{2+}$ and leaves the fluid (via precipitation of fluorite) and creates a negative Eu anomaly [59]. As a result, an acidic F-rich fluid migrating in the carbonate host rocks (regardless of other sources of Ca) dissolves $CaCO_3$ and forms fluorite (reaction III; [59,65]) with a low amount of $\Sigma$REE and a negative Eu anomaly in the initial stage of mineralization.

$$I. HF + F^+ \Leftrightarrow H^+ + 2F^-$$

$$II.\ H^+ + S^{2-} \Longleftrightarrow HS^-$$

$$III.\ CaCO_3 + 2H^+ + 2F^- \Longleftrightarrow CaF_2 + H_2O + CO_2$$

This explains the formation of the Pinavand purple fluorite, which has low $\Sigma$REE values and a negative Eu anomaly. The sulfide minerals (galena and pyrite) formed at this stage. The precipitation of purple fluorite decreased the pH and solubility of $REE^{3+}$. This yielded an increase in the $S^{2-}$ activity in the fluid, and consequently, the white fluorite with higher $\sum$REE values and a positive Eu anomaly was precipitated along with relatively more sulfides (Figure 4a).

### 6.6. Fluorite-Host Rock Relationship

The Eu/Eu* ratio in the host carbonate rocks of the Pinavand is 1.15–1.21, which shows a positive anomaly (Table 2). In the carbonate host rock, the Ce/Ce* ratio is 0.75–0.78 suggesting a negative Ce anomaly. The (La/Yb)n and (Tb/Yb)n ratios of the carbonate host rocks show high enrichment in LREE. In addition, the chondrite-normalized REE pattern (Figure 13) shows an enrichment in LREE in the carbonate host rock compared to the fluorite samples. Both purple and white fluorites, and the carbonates show negative Ce anomalies; however, the overall REE patterns are different in the fluorites and host rock. This may suggest that the REE content of the Pinavand fluorites has been partially provided by the carbonate host rock. The host rock could have been a major source of calcium for fluorite.

### 6.7. Source of REE and Hydrothermal Fluid

The micro-thermometry data for the purple and white fluorite samples show that overall, the ore-forming fluid of the Pinavand deposit had a moderate temperature and moderate to high salinity. It appears that two types of hydrothermal fluids were involved in the precipitation of fluorite. The first fluid had a higher temperature (~250 °C) and salinity (~35 wt% NaCl equiv.), which precipitated mainly purple fluorite (Figures 13 and 14). This fluid is likely related to an igneous source at depth and was enriched in LREE due to extreme evolution [71]. Although volcanic and plutonic rocks are not exposed at Pinavand, they do exist in the area (Figure 2). The high salinity of this fluid is likely related to the leaching of evaporites at depth during the ascending of the magmatic fluids.

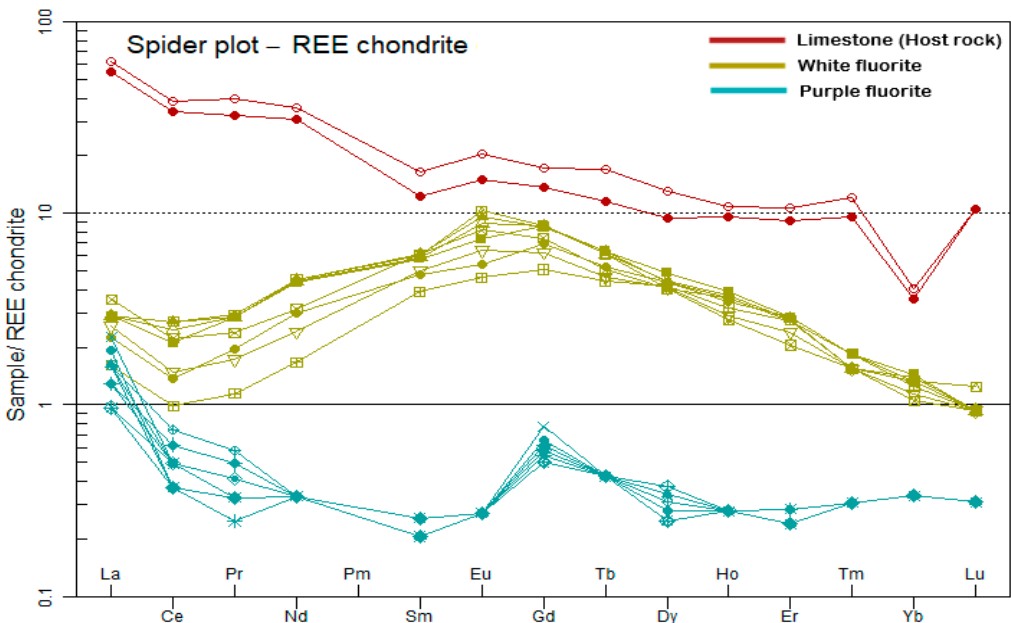

**Figure 13.** Chondrite-normalized REE patterns for fluorite and carbonate host rocks of the Pinavand deposit. The reference data for chondrites are from [70].

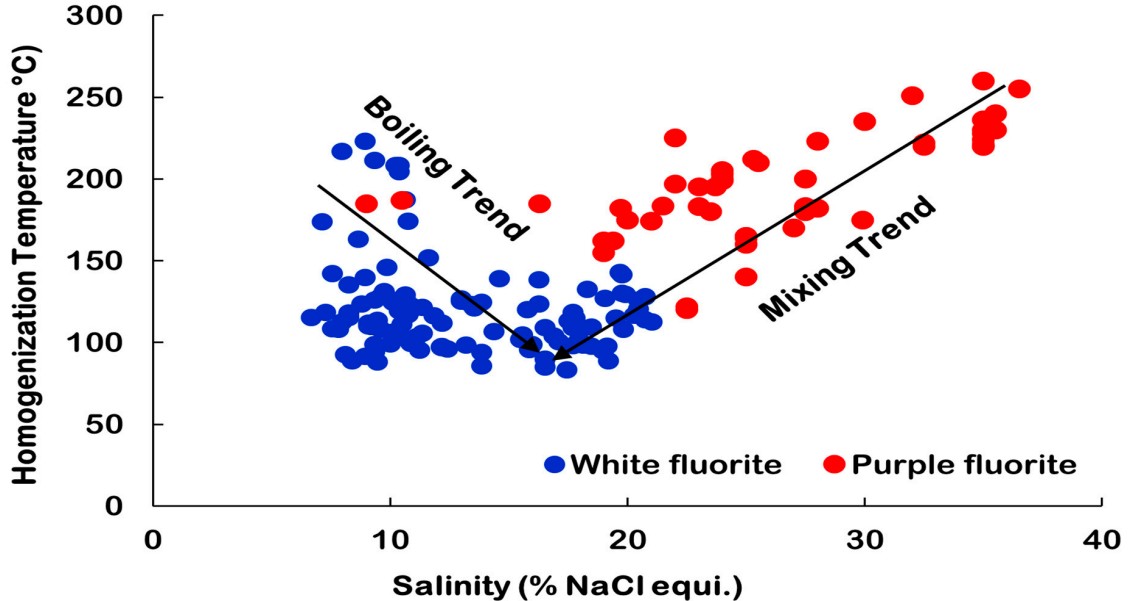

**Figure 14.** Homogenization temperature versus salinity diagram showing the evolution path of the hydrothermal fluids.

The second fluid had a lower temperature (~200 °C) and salinity (~10 wt% NaCl equiv.) (Figures 14 and 15). This fluid is likely from a meteoric source and evolved through boiling (Figure 14). Microthermometric data and the fluid evolution trends show mixing of the first and second fluids (Figure 14).

The fluid-rock interaction and the overall dissimilarity of REE patterns in fluorites and host rocks (Figure 13 and Table 2) show that the Cretaceous host rocks were not the major source of REE in the Pinavand deposit. In fluorite deposits, other possible sources of elements include siliciclastic host rocks (e.g., shales and sandstones), evaporitic rocks [72,73], and igneous intrusions [14,59,71,74]. In the Pinavand area, evaporite units are not exposed; however, Triassic shale and sandstone sequences are found locally (Figure 3); the meteoric water migrating through these sequences was enriched in REE and Y and

precipitated white fluorites (Figure 13). The igneous bodies in the area (Figure 2) are considered to be the source of the first hydrothermal solution, which was enriched in LREE (Figure 13).

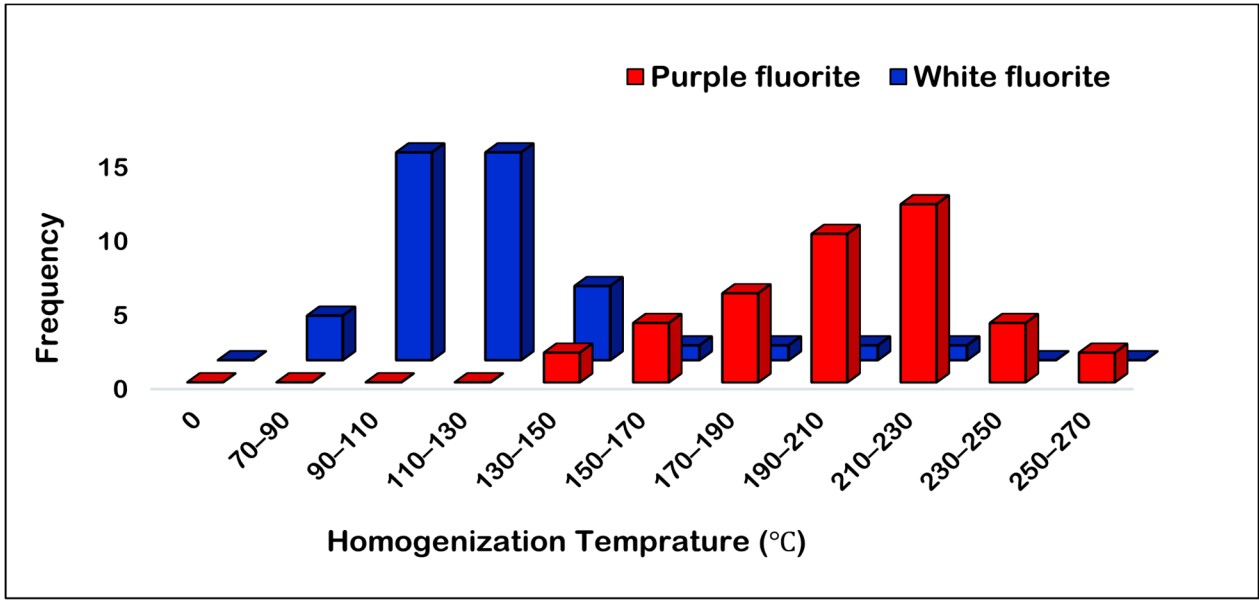

**Figure 15.** Histogram showing homogenization temperature of fluid inclusions in the Pinavand fluorites. Purple fluorite data from [18].

This fluid moved along suitable structures, such as local faults, to the site of deposition and precipitated the Pinavand purple fluorite due to reaction with the carbonate wall rock, cooling, and mixing with the second meteoric-sourced solution (Figures 14 and 16). Comparison of the Pinavand deposit with other fluorite-rich deposits (Table 5) indicates that the studied deposit in terms of genesis is similar to the Aguachile and Cuatro Palmas deposits, Coahuila, Mexico.

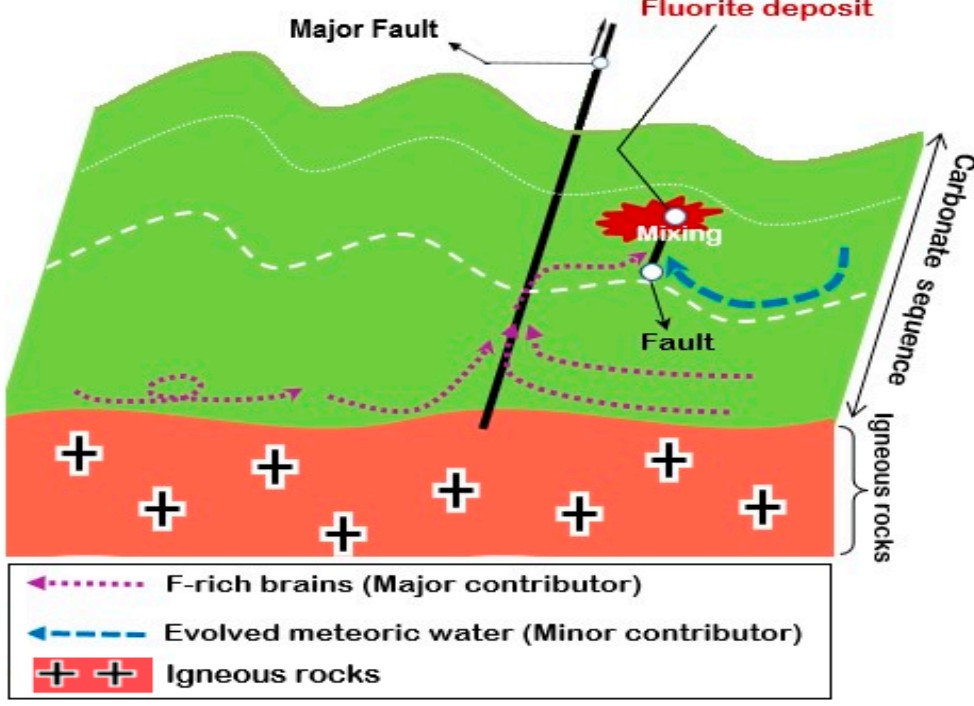

**Figure 16.** Schematic genetic model for the Pinavand fluorite deposit.

**Table 5.** Comparison of the Pinavand fluorite deposit with other fluorite-rich hydrothermal deposits (after [75]).

| Deposit | Type | Ore | Model of Formation | T (°C) | Salinity (wt% NaCl equiv.) | Comments | References |
|---|---|---|---|---|---|---|---|
| Las Cuevas & Río Verde, San Luis Potosí | Skarn-related | Fluorite | Contact metamorphism and retrograde hydrothermal fluids from F-rich volcanic rocks. | 60 to 130 | ~0 | | [76] |
| | MVT | | Diluted basinal brines reacted with host limestones. | 60 to 110 | 0 to 0.2 | | [77] |
| | MVT-like | | Diluted basinal brines reacted with F-rich modified meteoric water and host limestones. | 49 to 177 | 0 to 1.9 (mostly 0.2) | | [75] |
| Bolaños & San Martín de Bolaños, Jalisco | Epithermal | Polymetallic, rich in fluorite | Essentially Ag-rich intermediate sulfide deposits with stages of mineralization very rich in fluorite, deposited through boiling or conductive cooling. | 150 to 340 | 0 to 16 | | [78,79] |
| Several small deposits in Central Mexico | Tin rhyolites | Sn | Fumarolic deposits associated with extremely differentiated F-rich rhyolites and rhyodacites. | n.a. | n.a. | | [80,81] |
| Buenavista, Coahuila | MVT | Fluorite | Dense F-rich basinal brines that reacted with platform and reef carbonates. | 50 to 155 | 5.7 to 18.1 | | [75,82–84] |
| El Pilote, Coahuila | Skarn | Fluorite | Shallow hypabyssal rocks associated with hydrothermal fluids that dissolved pre-existing MVT-like fluorite mantos, and fluorite re-precipitated around the skarn. | 78 to 423 | 5 to 34 | 24.5 to 29.1 wt% $CaCl_2$ fluids | [75,85–87] |
| Aguachile and Cuatro Palmas, Coahuila | Shallow hydrothermal | Fluorite, Be, U, Mo, etc. | Fluids largely exsolved from cooling hypabyssal alkaline to calc-alkaline rocks that reacted with host carbonate rocks. | 70 to 180 | 0.9 to 8.8 | Fluids generally below 4 wt% NaCl equivalents | [75,88,89] |
| Pinavand district, Iran | Hydrothermal | Fluorite | Reaction of the hydrothermal solution. with host limestone | 75 to 270 | 0.3 to 36 | | |

MVT = Mississippi Valley Type; n.a. = not available.

*6.8. Source(s) of Sulfur*

The $\delta^{34}$S values of the barite samples are +21.1‰ and +25.4‰ (Table 4, Figure 17). The $\delta^{34}$S values of galena are in the range of −0.2‰ to −3.7‰. Comparison of $\delta^{34}$S values of the studied samples with isotopic values of seawater sulfate, determined by Claypool et al. [90], shows that $\delta^{34}$S values of the Pinavand barite samples are heavier than seawater sulfate contemporaneous with its host rock (Cretaceous) and are similar to seawater sulfate in the upper Proterozoic (Figure 17). It is possible that the brines trapped in the upper Proterozoic rocks precipitated the Pinavand barites. Low values of $\delta^{34}$S in galena suggest that the process of bacterial sulfate reduction (BRS) may be the most important mechanism of sulfur production.

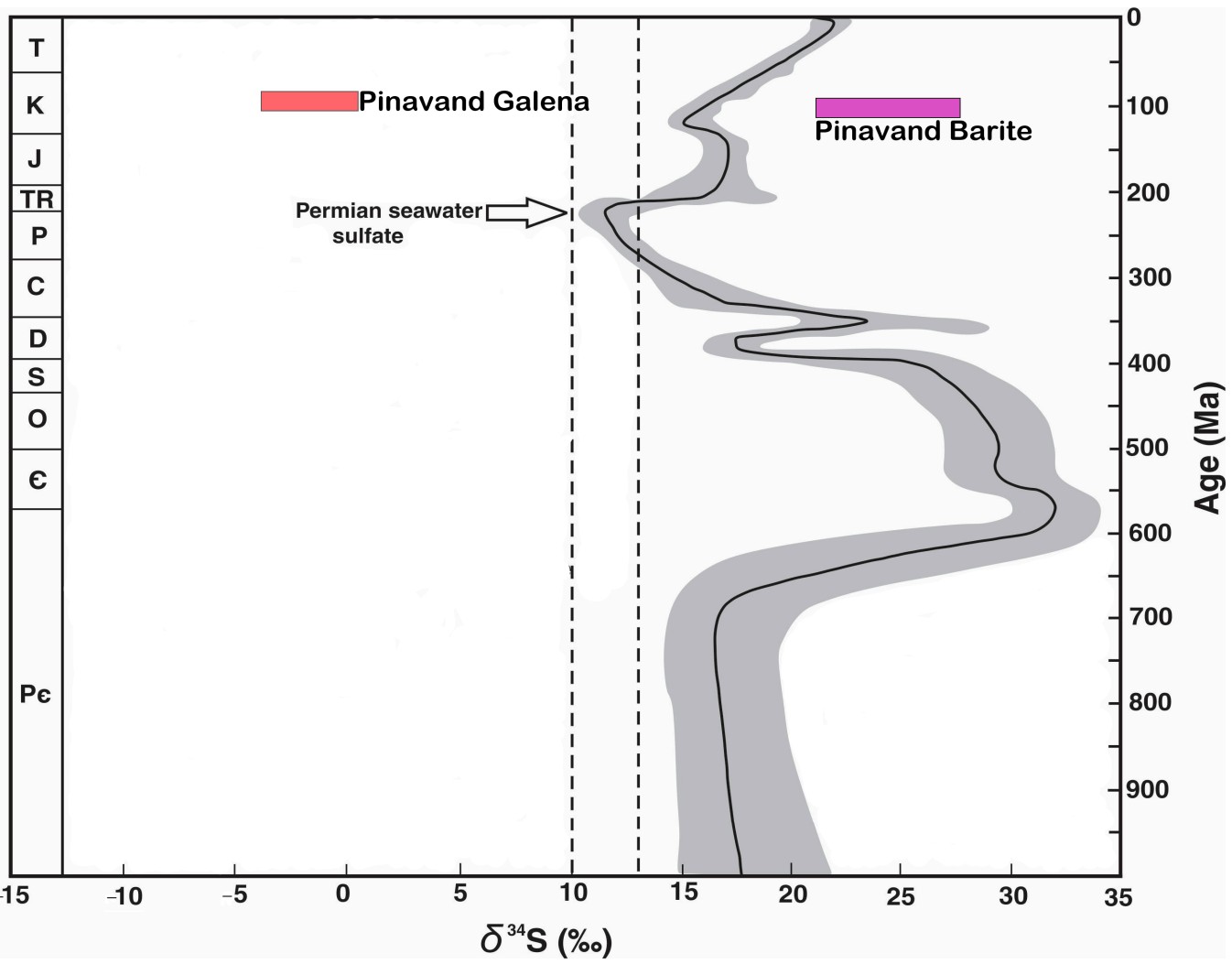

**Figure 17.** Distribution of $\delta^{34}$S values in sulfate and sulfide minerals of the Pinavand deposit in relation to the age curve for sulfur (boundaries after [90]).

Two processes can cause sulfate reduction and the production of reduced sulfur for the formation of sulfide minerals: thermochemical sulfate reduction (TSR) and BSR [91–93]. In TSR, which is a non-biological slow process [94], sulfate is reduced to sulfur under the influence of therm [95], which is mostly effective in the temperature range of 100 °C to 140 °C. The reduced sulfur created as a result of this process shows 0‰ to 15‰ depletion compared to the primary sulfates [96].

Temperature is one of the key factors limiting the direct involvement of BSR in biogenic sulfide deposition. It is well known that microorganisms can live at considerable depths [97,98]. Kucha et al. [98] stated that subsurface environments can support high densities of microbes within the pore network realm, provided the temperature remains below 120 °C. It has also been found that sulfate-reducing bacteria can live in sorely saline environments (up to about 20 wt% NaCl [99]) or even up to the halite saturation stage [93]. However, the optimal conditions of sedimentation are in ordinary salinity solutions (10–25% NaCl equiv.) [99]. Most thermophilic microorganisms are susceptible to temperature changes, and their optimal living conditions are much lower than 120 °C [97].

## 7. Conclusions

The Pinavand fluorite deposit, as veins and replacement masses, formed in the carbonate host rocks near the UDMA magmatic belt and the SSZ metamorphic zone. In the Pinavand deposit, the fluorite veins show a zoning where dark to light purple fluorites are gradually replaced by white to smoky or cream fluorites. Two distinct generations of fluorite were precipitated: early-stage high-temperature, high-salinity purple fluorite with low REE and Y, and late-stage low-temperature, low-salinity white fluorite with higher REE and Y (Figures 4 and 5).

The La/Ho, La/Yb, and Sr values are higher in the white fluorite, whereas the Tb/Yb and Y/Ho ratios are higher in the purple variety. The carbonate host rock has a higher La/Yb than the fluorite samples. This host rock and the purple fluorite show enrichment in LREE, whereas the white fluorite is enriched in HREE. Differences in the REE pattern and their concentration in both fluorite types and the host rock suggest that the carbonate country rocks did not have a significant role as the REE source. The LREE-enriched fluid is considered to be related to the extensive magmatism that occurred in the region, although its products are not exposed at the Pinavand. However, the REE ratios and concentrations were modified due to the interaction of the magmatic fluid with the carbonate wall rocks, mixing with meteoric water, and changes in the physico–chemical conditions of the mineralizing system (pH, T, $fO_2$). Comparison of $\delta^{34}S$ values of the Pinavand barites (+21.1‰ to +25.4‰) with isotopic values of seawater sulfate shows that the Pinavand barites values are heavier than those in the Cretaceous limestone seawater sulfate but are similar to those in the upper Proterozoic seawater sulfate. The low $\delta^{34}S$ values in galena (−0.2‰ to −3.7‰) suggest BRS process probably had a controlling effect on sulfur production.

It seems that two main fluids were responsible for the mineralization in the studied deposits (Figure 16). A high-temperature, high-salinity fluid was possibly related to an igneous source at depth. This fluid has moved along suitable structures, such as local faults, to the place of mineral deposition. The high salinity of this fluid is probably related to the washing of evaporites at depth during the ascent of magmatic fluids. This solution could also contain metal complexes. The second fluid had a lower temperature and salinity (Figures 14, 15 and 18). This fluid probably had a meteoric source (Figure 16) that migrated through the sandstone and shale sequences and contained sulfate from a surface origin that was associated with seawater sulfate ions. The first fluid precipitated minerals during reaction with the carbonate wall rock, cooling and mixing with the second solution of meteoric origin (Figures 14 and 16). Temperature changes were the result of continuous mixing of high-temperature and low-temperature mineralizing fluids with different ratios at shallow depths during deposition. This mixing of fluids maintained the final habitability conditions of the endothermic bacteria (Figure 18) [100].

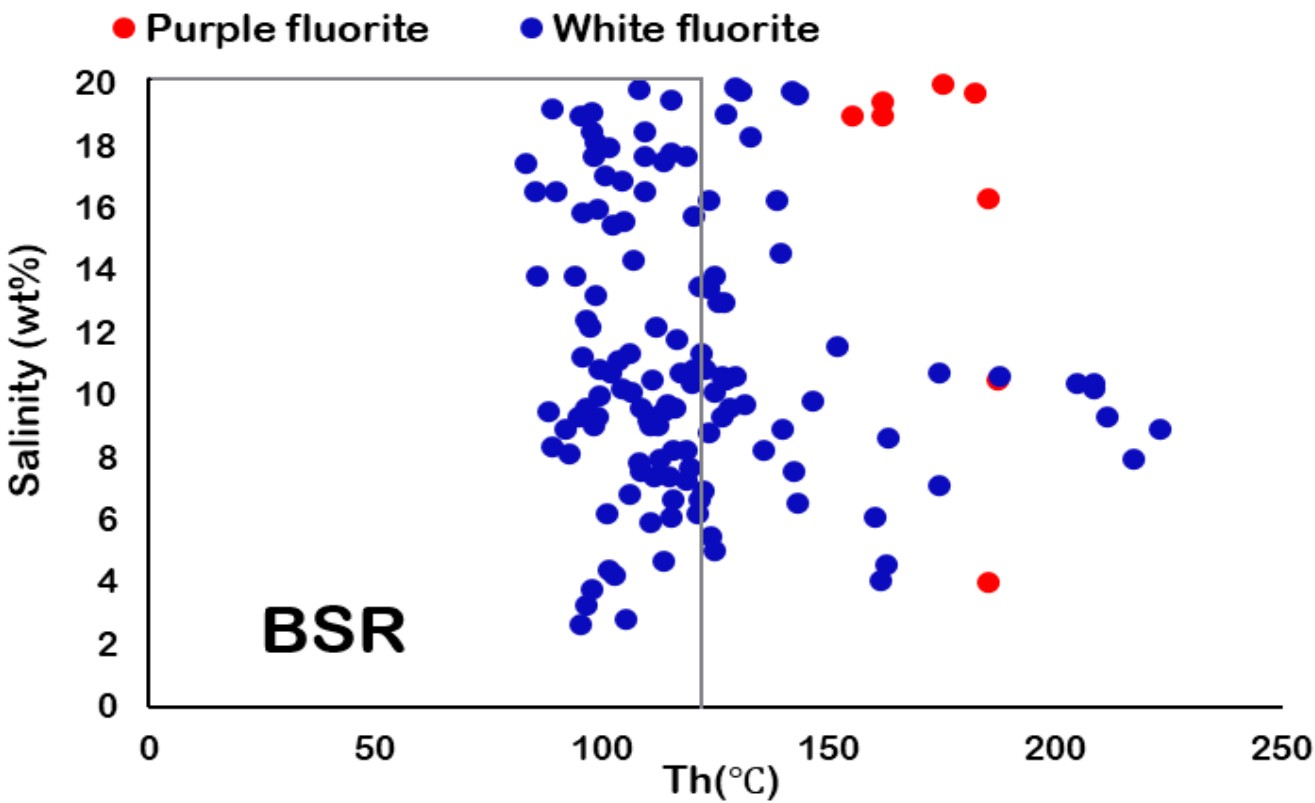

**Figure 18.** Salinity-temperature diagram for the studied deposit.

**Author Contributions:** F.G., B.T., A.K.S. and S.F. conceived and designed the experiments; F.G., B.T., A.K.S. and S.F. provided reagents/materials/analysis tools; F.G., B.T., A.K.S. and S.F. wrote the paper. All authors have read and agreed to the published version of the manuscript.

**Funding:** This research was funded by research grant from the Shiraz University, Iran.

**Data Availability Statement:** Not applicable.

**Acknowledgments:** This project has been supported by the research committee of Shiraz University, Iran. Many thanks to Farhad Ahmadnejad for the fluid inclusion analyses. We are grateful to Farid Moore for his valuable and constructive comments. Many thanks to two announce reviewers for their constructive comments to improve the manuscript.

**Conflicts of Interest:** The authors declare no conflict of interest.

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
