# Peer review of "Fluid Inclusions and REE Geochemistry of White and Purple Fluorite: Implications for Physico-Chemical Conditions of Mineralization; an Example from the Pinavand F Deposit, Central Iran"

_minerals, doi:10.3390/min13070836_

Round 1

Reviewer 1 Report

This study presents fluid inclusion data, REE analysis (ICP-OES and ICP-MS) as well as sulfur isotope analysis on fluorite from Iran.  Two fluorite varieties are compared: purple fluorite vs. white fluorite. Both fluorite varieties could be differentiated based on their geochemical characteristics.

The authors conclude that the conditions of fluorite mineralization must have changed from low to high oxygen fugacity  and  from low to high pH during subsequent lower temperatures. The purple fluorite must have formed earlier at higher temperatures (argumentation however can be improved, see below). Two main mineralizing fluid systems can be distinguished: an earlier high-temperature, high-salinity fluid (salt from evaporates), which was possibly related to an igneous source at depth and a low-temperature fluid. Sulfur isotopes show that biogenic sulfate reduction was important during the last stage of mineralization.

REE patterns have been compared with other fluorite deposits.

The quality of the data presented here is generally good and allow some interesting conclusions considering the mineralizing conditions of fluorite deposition.

The paper is suitable for publication, but shows several shortcomings, which must be resolved before publication.

One of the main critics consider:

-        - The sample material of purple fluorite used for fluid inclusion studies. The authors state that the purple fluorite from Pinavand does not contain measurable fluid inclusions. Therefore, microthermometry data of another deposit (Qishlaqi) were used here. However, it is not clear how fluorite from the latter locality relate to the white fluorite from Pinavand. It is not clear either, which samples were used for REE analysis (Pinavand or Quishlaqi, or both).

-        - Lack of petrographic information. The geochemical data point at different stages of fluorite mineralization (early purple fluorite and late white fluorite). However, it is not explained how these fluorite varieties occur in the field and if age difference is supported by structural evidence.

-      -   The microthermometry data would be more complete by giving also the eutectic temperatures. That information would show if the solutions contain NaCl, or other salts (CaCl2?)

-      -   Interpretation: The authors consider two trends of fluid evolution (Fig. 12): fluid mixing (purple fluorite) and boiling (white fluorite). However, the temperatures recorded here are based on samples from different localities. Furthermore, “boiling” seems to be not supported by fluid inclusion-related micro-textures (e.g. heterogeneous trapping) and questionable (isothermal mixing would fit as well). It is not clear either if we have really to do with only two stages, an early stage (purple fluorite) and a late stage (white fluorite). Perhaps more stages of fluid activity played a role. Again, some more details about the petrography and notably the relation between both fluorite varieties would help.

Corrections and comments are written in the uploaded PDF file.

Main comments and suggestions follow below:

Change the title: Replace “Microthermometry” by “Fluid inclusions”. Fluid inclusions sounds better in the title. The information obtained from fluid inclusions is more than microthermometry.

128-129: …. However, microthermometric data of purple fluorite from Qishlaqi [18] is used here for comparison …. : It should be better explained why the use of these data is justified. What is the relation between Qishlaqi and the present deposit?

Table 3: The accuracies (decimals) for salinity and inclusion size are not realistic. The salinity cannot be calculated with 0.01 wt% accuracy from Tm Ice (with accuracy in the order of 0.5°C).

l.145      The paragraph “5.1. Mineralization and textural relationship” does not belong to “Results” Not a very good text. Information is all mixed. The geological information fits better in the Geological Setting.

What is missing: the relation between white and purple fluorite. In this paper, you compare these two fluorite varieties (i.e. based on the color). However, the geological /structural correlation between the two is not clear. Do they occur in the same hand specimen? What about the purple fluorite from Quishlaqi?

l.187: … however they range from vapor-rich to liquid-rich varieties without thermometric differences….. What do you mean here? Inclusions with different water volume fractions show different Th? That would be not possible.

… purple fluorites ranges from 90 to 150 °C and 170 to 270 °C, respectively…. : Which fluid inclusions are meant here? Inclusions from Quishlaqi?

l.191: … The salinity of fluid inclusions in purple fluorite (2.5 - 36 wt% NaCl equiv) are higher than that in the white fluorite (2.6 - 20.43 wt% NaCl equiv)… : Actually, in both fluorites the salinity of the fluid inclusions shows extreme ranges. However, in Fig. 4 it is shown that purple fluorite has actually salinity > ca. 18 wt%, whereas the lower values are very rare. Better give the ranges where most salinities plot. That is clearer. The 2.5 wt% NaCl-eq. inclusion seems to be an exception. 

Did you measure eutectic temperatures? That would show if the solution contains NaCl or perhaps other salts, like CaCl2!

l.208: …that both fluorite types were precipitated from two chemically distinct hydrothermal solutions…. : It was mentioned above that fluid inclusions were measured in different purple fluorite as the fluorite from Pinavand does not contain fluid inclusions. Which fluorite was taken here for REE analysis?

337:  ….The first fluid had higher temperature (~250 °C) and salinity (~35 wt% NaCl equiv.) which precipitated mainly purple fluorite …..: The purple fluorite is older, based on geochemical data. However, does this conclusion concur with petrological observation?

410: …Two distinct generations of fluorite were precipitated…. Here, you speak of two generations of fluorite for the first time. Again, is there any petrological evidence which supports this assumption?

The English is reasonable, but can be improved.

Author Response

Manuscript ID: minerals-2381392

Title: Fluid inclusions and REE geochemistry of white and purple fluorite: implications for physico-chemical conditions of mineralization; an example from the Pinavand F deposit, Central Iran.

Authors: Fatemeh Ghaedi, Batoul Taghipour *, Alireza Somarin, Sa Reviewers maneh Fazli

Dear Editor and Reviewers,

All comments of editor and reviewers were accepted and modified in the revised manuscript. The revised parts of the manuscript that follow the suggestions and questions by editor and reviewers are as follows in red color in the main text.

Pages number and lines are based on the manuscript revised file:

Reply to Reviewer 1

  1- The sample material of purple fluorite used for fluid inclusion studies. The authors state that the purple fluorite from Pinavand does not contain measurable fluid inclusions. Therefore, microthermometry data of another deposit (Qishlaqi) were used here. However, it is not clear how fluorite from the latter locality relate to the white fluorite from Pinavand. It is not clear either, which samples were used for REE analysis (Pinavand or Quishlaqi, or both).

Thank you. As mentioned in the text, the studied purple fluorites were fine and not suitable for measuring fluid inclusions. Therefore, the fluid inclusion data of the purple fluorites which studied by Qishlaqi in the Pinavand area, used for comparison with white fluorites data. Both purple (Previous studied by Qishlaqi, 2002) and white fluorites (this study) are belong to the fluorite sample of the Pinavand mining district. The REE data of white and purple fluorites using in the section of rare earth elements geochemistry, both were carried out in this research (ACME laboratory, Canada) (Please see Section 4 sampling and analytical methods, page9):

“Mineralogy, geochemistry, and microthermometry studies were conducted on 27 representative samples collected from the tailings and outcrops of the Pinavand deposit. The chemical analysis of the present study was carried out on different generations of fluorite, 8 white and 12 purple fluorite samples in ACME laboratory, Canada and provided in Tables 1, 2.”

 “Therefore, the purple fluorites previously studied in the Pinavand deposit by Qishlaqi [18] with a higher frequency in the temperature range of 170 to 260 ℃ were used to compare with the white fluorites in this study with a temperature range of 90 to 150 ℃.”

2- Lack of petrographic information. The geochemical data point at different stages of fluorite mineralization (early purple fluorite and late white fluorite). However, it is not explained how these fluorite varieties occur in the field and if age difference is supported by structural evidence.

Thanks. Field and petrographic images of the Pinavand fluorite types were added in the text, Mineralization and textural relationship section 3.1 (Figures 4 &5).  

 “In the Pinavand region, the veins of fluorites show a zoning, dark to light purple fluorites are gradually replaced by white to smoky or cream fluorites indicate the fluorite mineralization occurred in two stage and this zoning may have been created by the crushing or complete dissolution of the older generation (Figure 4h).

In the region of the Pinavand, there is a distinct variety of textures and unique among which can mention radial-fibrous, pseudo-acicular and plumose textures (Figure 4g; 5a, c). These textures are created as a result of fluid circulation within fractures and also due to processes such as boiling (mostly in epithermal deposits) [95, 96, 97]. In the Pinavand region, these textures have been observed in barite minerals formed during the mineralization stages of purple and white fluorites (Figure 5a, c, d).” and “Figures 4&5”

3 - Did you measure eutectic temperatures? That would show if the solution contains NaCl or perhaps other salts, like CaCl2!

Thank you. The eutectic temperature is so helpful, but the laboratory did not measure the Te. Qishlaqi (2002) has been shown that Pinavand as a multi-component system and includes NaCl, CaCl2 and MgCl2.

 4 - Interpretation: The authors consider two trends of fluid evolution (Fig. 12): fluid mixing (purple fluorite) and boiling (white fluorite). However, the temperatures recorded here are based on samples from different localities. Furthermore, “boiling” seems to be not supported by fluid inclusion-related micro-textures (e.g., heterogeneous trapping) and questionable (isothermal mixing would fit as well). It is not clear either if we have really to do with only two stages, an early stage (purple fluorite) and a late stage (white fluorite). Perhaps more stages of fluid activity played a role. Again, some more details about the petrography and notably the relation between both fluorite varieties would help.

In this regard, both textural evidence and fluid inclusions data support boiling process. Therefore, images related to boiling were added with the necessary explanations in the paper as follows, please see page 6 section 3.1: Mineralization and textural relationship & Section 5.2: Fluid inclusion microthermometry:

 “Coexisting liquid-rich and vapor-rich fluid inclusions contained in the Pinavand fluid inclusion assemblage [99] (Figure 6c) provide indisputable proof of the attendance of boiling fluids [98,100] (Figure 6c). and “Figure 6”.

Thank you. we understand the reviewer's concern regarding the structure of the paper, we would like to point out that purple and white fluorites have been investigated in at least two generations and are the main focus of this project. In order to confirm stages of fluid activity, there is a probability of the existence of more stages of mineralization of fluorites in the Pinavand region based on the color variation, petrographic evidence and the relationship of fluorites, this needs more analysis of all types of fluorites.

 5- Change the title: Replace “Microthermometry” by “Fluid inclusions”. Fluid inclusions sounds better in the title. The information obtained from fluid inclusions is more than microthermometry.

Thanks, as you suggested microthermometry changed by fluid inclusion.

6 - 128-129: …. However, microthermometric data of purple fluorite from Qishlaqi [18] is used here for comparison ….: It should be better explained why the use of these data is justified. What is the relation between Qishlaqi and the present deposit?

Qishlaqi (2002) studied the Pinavand fluorite mineralization. We using and comparised the Qishlaqi data in this study.

7 - Accuracy of 0.1 μm size sounds ridiculous. 

We revised Table 3, the size of fluids is 5-20.

8 - Table 3: The accuracies (decimals) for salinity and inclusion size are not realistic. The salinity cannot be calculated with 0.01 wt% accuracy from Tm Ice (with accuracy in the order of 0.5°C).

Thank you.  As you suggested it revised as decimals in 0.1 wt%. Please see Table 3.

9 - l.145 The paragraph “5.1. Mineralization and textural relationship” does not belong to “Results” Not a very good text. Information is all mixed. The geological information fits better in the Geological Setting.

Thanks. We revised text, please see Section 3.1 (Mineralization and textural relationship). Page 6.

10 - What is missing: the relation between white and purple fluorite. In this paper, you compare these two fluorite varieties (i.e., based on the color). However, the geological /structural correlation between the two is not clear. Do they occur in the same hand specimen? What about the purple fluorite from Quishlaqi?

The white and purple fluorites have been observed both separately and simultaneously in the same sample in the Pinavand area. In many examples that have been seen next to each other, they display a color zoning, which indicates the different generations of fluorites as well as Qishlaqi (2002). Hence, some images were added and reported in the paper.

11 - l.187: … however they range from vapor-rich to liquid-rich varieties without thermometric differences…. What do you mean here? Inclusions with different water volume fractions show different Th? That would be not possible.

Thanks We revised this sentence.

Section 5.2 (Fluid inclusion microthermometry): however, they range from vapor-rich to liquid-rich varieties.

12 - … purple fluorites ranges from 90 to 150 °C and 170 to 270 °C, respectively….: Which fluid inclusions are meant here? Inclusions from Quishlaqi?

The purple fluorites with a temperature range of 170 to 270 ℃ are from Qishlaqi (2002) and the white fluorites with a temperature range of 90 to 150 ℃ (present study).

13- l.191: … The salinity of fluid inclusions in purple fluorite (2.5 - 36 wt% NaCl equiv) are higher than that in the white fluorite (2.6 - 20.43 wt% NaCl equiv) …: Actually, in both fluorites the salinity of the fluid inclusions shows extreme ranges. However, in Fig. 4 it is shown that purple fluorite has actually salinity > ca. 18 wt%, whereas the lower values are very rare. Better give the ranges where most salinities plot. That is clearer. The 2.5 wt% NaCl-eq. inclusion seems to be an exception. 

 Thanks. Revised this figure and deleted very low salinity.

14 - The microthermometry data would be more complete by giving also the eutectic temperatures. That information would show if the solutions contain NaCl, or other salts (CaCl2?)

Thank you. As explained in comment 3, Qishlaqi (2002) has been shown that Pinavand as a multi-component system and includes NaCl, CaCl2 and MgCl2.

15 - 337:  …. The first fluid had higher temperature (~250 °C) and salinity (~35 wt% NaCl equiv.) which precipitated mainly purple fluorite ….: The purple fluorite is older, based on geochemical data. However, does this conclusion concur with petrological observation?

Thank, this conclusion is consistent with petrological observations; On the based geological and textural evidence, the purple and white fluorite occurred in the dolomitic host rocks, there is distinct relationship between dolomitic host rock, fluorite, barite and galena mineral association so pictures were added (Please see Figure 4). This issue was discussed in previous the comment 2.

16 - Deutsch (Deutschland)

Thanks. We checked this reference it is as the following:

Reference number 11: Möller P, Maus H, Gundlach H (1982) Die Entwicklung von Flußspatmineralisationen im Bereich des Schwarzwaldes. Jahresh Geol Landesamt Baden-Wuerttemberg 24: 35–70.

Reviewer 2 Report

The paper is interesting and valuable, however some corrections are recommended:

1.      Geological setting description (line 56 -80) is very general and not compatible with fig. 1 legend.  May be simplified by description  only of zones were fluorite deposits are located supported by fig 2a.

2.      Description of deposit geology (line 144 166 chapter 5.1: “Mineralization and textural relationship”) should be located before sampling and analytical methods and combined with chapter 3 : “Geology of Pinavand area” . Addition of cross section through studied deposit is recommended.

3.      On Fig. 2 b geology of Ardestan and Sejzi Bains  (marked white) is not explained

4.      The studied are two fluorite types (white and purple) but there is no description of the mode of occurrence of both and their mutual relationship (in replacement bodies and veins) They appear separately? Where. The mode of occurrence of both type should be described and addition of corresponding illustration (photographs, designs) is recommended.

5.      The description of Eu and Ce anomalies is not clear. Negative Eu/Eu* anomaly suggesting low fO2 in purple fluorite and negative Ce/Ce* suggesting high fO2 needs a comment.

6.      The conclusions should be supported by data on spatial distribution and mode of occurrence of purple and white fluorite.

Author Response

Manuscript ID: minerals-2381392

Title: Fluid inclusions and REE geochemistry of white and purple fluorite: implications for physico-chemical conditions of mineralization; an example from the Pinavand F deposit, Central Iran.

Authors: Fatemeh Ghaedi, Batoul Taghipour *, Alireza Somarin, Sa Reviewers maneh Fazli

Dear Editor and Reviewers,

All comments of editor and reviewers were accepted and modified in the revised manuscript. The revised parts of the manuscript that follow the suggestions and questions by editor and reviewers are as follows in red color in the main text.

Pages number and lines are based on the manuscript revised file:

1- Geological setting description (line 56 -80) is very general and not compatible with fig. 1 legend.  May be simplified by description only of zones were fluorite deposits are located supported by fig 2a.

Thanks. The Pinavand area is located in the Central Iranian Block. The geological setting of Central Iranian Block was described in the “Geological Setting” section and is shown by figure 2a.

2 - Description of deposit geology (line 144 166 chapter 5.1: “Mineralization and textural relationship”) should be located before sampling and analytical methods and combined with chapter 3: “Geology of Pinavand area”. Addition of cross section through studied deposit is recommended.

Thank you. We revised text. Please see page 6, lines 111-140.  Section 3.1 (Mineralization and textural relationship).

 3- On Fig. 2 b geology of Ardestan and Sejzi Basins (marked white) is not explained.

Figure 2 caption: Ardestan and Seizi Basins are sedimentary basins in the Central Iranian zone [31].

4 - The studied are two fluorite types (white and purple) but there is no description of the mode of occurrence of both and their mutual relationship (in replacement bodies and veins) They appear separately? Where. The mode of occurrence of both types should be described and addition of corresponding illustration (photographs, designs) is recommended.

Thank you. We revised the text. Please see page 6, lines 131-140. Figures 4 & 5.

5 - The description of Eu and Ce anomalies is not clear. Negative Eu/Eu* anomaly suggesting low fO2 in purple fluorite and negative Ce/Ce* suggesting high fOneeds a comment.

The reviewer’s accuracy is commendable. It was reviewed: It is assumed that oxygen fugacity was completely variable during mineralization and this rate was high during the formation of fluorites with negative Eu and Ce anomalies. Also, since this is not the only influencing factor in precipitation and Ce anomaly has been continuously negative, so this case cannot be justified by the oxygen fugacity factor alone. Therefore, we justify this case in another way. The following changes were made for clarification and more appropriate justification:

Section 6.3 (6.3. Eu and Ce anomalies): The negative Eu anomaly reflects presence of Eu2+ (instead of Eu3+) in the hydrothermal solution during purple fluorite deposition [47, 14]. As a result, at a temperature above 200 ℃, due to the radius difference, Ca2+ has not been replaced by Eu2+. Therefore, Eu depletion is observed in fluorites [46, 5, 57, 13]. Such a solution precipitates fluorite as temperature decreases or fO2 increases [55, 5]. Fluid inclusions data in the Pinavand area indicated purple fluorites are formed at high temperatures (i.e., >200 ℃). Co-precipitation of fluorite with other Eu-enriched or depleted minerals may also produce positive or negative anomalies [53]. The simultaneous observation of enrichment and depletion in Eu in fluorites of the same region indicates the existence of two mixed fluids with different temperatures and amounts of Eu [12].”

“All fluorite samples of the Pinavand deposit show a negative Ce anomaly (i.e., Ce/Ce*<1) from 0.38 to 0.85. The persistent negative Ce anomaly in the Pinavand fluorites may be due to: 1) the oxidation of the solutions at the source [46], so after that Ce3+ oxidation and Ce4+ immobilization [46, 40], 2) the depletion of Ce can be the result of their formation from a reduced fluid the presence of sulfide minerals in Pinavand area can indicated reduced condition [49] (Figure 4a), 3) may also be related to hydroxide complexation [57]; the hydroxide complex formed by Ce is more stable than other REEs [58], which causes Ce to remain in the fluid and, consequently, a negative Ce anomaly is observed in the precipitants [53].”

6 - The conclusions should be supported by data on spatial distribution and mode of occurrence of purple and white fluorite.

Thank you. Revised the text, please see page 26, lines 459-461.

Round 2

Reviewer 1 Report

The second revised version of the manuscript has been improved compared to the first version. However, there are still some points which can be considered before publication.

- The use of literature data of fluid inclusions from the purple fluorite is OK, but for me it is still not clear if this is the same sample material as decribed in the petrography. Evidentally this fluorite is free of fluid inclusions. Is it really the same? Why do you think so (only the colour cannot be reason).

- No information on intitial (eutectic) melting. It is still not clear if the solute if NaCl is the main salt in solution. It is likely that the solutes contain NaCl, but do you have any indications? Particularly in the high-salinty inclusions eutectix melting must be easily observed.

- The salinity in wit% given in 2 decimals is not realistic, as mentioned in the first report. This has been changed now, but strangely you added again 2 decimals in the inserted text. (l.223).

. English: Please take care of the following:

l.37 hance..... ???? =hence

l..54 understating... This word does not make sense here. =understanding?

l. 101: .3.1. (delete one point).

Author Response

Dear Reviewer,

Thank you for your constructive comments and very kind attention.

- The use of literature data of fluid inclusions from the purple fluorite is OK, but for me it is still not clear if this is the same sample material as decribed in the petrography. Evidentally this fluorite is free of fluid inclusions. Is it really the same? Why do you think so (only the colour cannot be reason).

Thanks, you're right.  Our samples are different from the Qishlaqi (2002) samples. In this research we focused on the color of fluorite as well as geochemical and fluid inclusions data indicating differences between two types of color fluorite (Figures 9, 10, 11, 12, 14). Your idea is fantastic and it should be considering the other geological and mineralogical factors to improve this research for next work.

- No information on intitial (eutectic) melting. It is still not clear if the solute if NaCl is the main salt in solution. It is likely that the solutes contain NaCl, but do you have any indications? Particularly in the high-salinty inclusions eutectix melting must be easily observed.

Thanks. As we described in the previous revision, unfortunately, we haven’t any data for initial melting (Te). However, according to Qishlaqi (2002), the present of halite phase in fluid inclusions which indicated NaCl is the main salt, also the eutectic temperature (Te= -23 to -57 ͦC) was determined, which indicated the presence of Ca, Na and Mg solutes in fluid inclusions (Qishlaqi, 2002), which can be evidence to prove the presence of other solutes in the fluid inclusions in the Pinavand region.

- The salinity in wit% given in 2 decimals is not realistic, as mentioned in the first report. This has been changed now, but strangely you added again 2 decimals in the inserted text. (l.223).

Thank you, the text was revised.

Comments on the Quality of English Language

. English: Please take care of the following:

l.37 hance..... ???? =hence

l..54 understating... This word does not make sense here. =understanding?

  1. 101: .3.1. (delete one point).

Thanks a lot for your detailed revisions, all mistakes were revised.

Kind regards,

Batoul Taghipour
